



# Catastrophic beach erosion induced by littoral drift on nearby beach after Samcheok LNG's massive coastal reclamation project

Changbin Lim[1], Tae Min Lim[2], Jung-Lyul Lee[3]

[1]IHCantabria - Instituto de Hidráulica Ambiental, Universidad de Cantabria, Santander, 39011, Spain
[2]School of Civil, Architectural Engineering & Landscape Architecture, Sungkyunkwan University, Suwon 16419, Republic of Korea
[3]School of Water Resources, Sungkyunkwan University, Suwon 16419, Republic of Korea

*Correspondence to*: Jung-Lyul Lee (jllee6359@hanmail.net)

**Abstract.** Large-scale construction projects, such as port construction and reclamation endeavors, can alter inshore wave
dynamics, leading to severe coastal erosion. In South Korea, recent large-scale reclamation projects have resulted in the catastrophic erosion of sand along nearby coastlines. This study focuses on Wolcheon Beach, where the complete sand loss occurred due to robust longshore sediment transport (LST) induced by a reclamation project for the construction of the nearby Samcheok liquefied natural gas (LNG) terminal in Gangwon Province. A shoreline change model was employed to simulate this phenomenon, and the results were validated using satellite imagery. The model accuracy was assessed by comparing the
LST rate vectors indirectly estimated from the changes in the shoreline delineated in satellite images with those directly derived from the model. Furthermore, a response methodology was proposed using the parabolic bay shape equation, which can effectively mitigate coastal erosion by controlling LST by installing a small-scale groin group on the adjacent beach before commencing reclamation or port projects. These findings are anticipated to significantly contribute to averting catastrophic coastal erosion issues, such as those witnessed at Wolcheon Beach, before performing large-scale construction in coastal
regions.

## 1 Introduction

Coastal areas are home to nearly half of the global population, and they are much more densely populated compared to inland areas although coastal areas occupy a small fraction of the Earth's surface. Approximately two-thirds of the world's megacities are situated 60 km from shorelines, boasting immense economic significance for marine transportation, fishing, and tourism
(Neumann et al., 2015). Ports facilitate 80 to 90 % of global trade, and many individuals choose coastal regions as vacation destinations (Notteboom and Rodrigue, 2005). Coastal zones are increasingly serving as leisure and cultural hubs, making effective coastal management imperative (OECD, 2007). Although coastal development provides economic benefits, it can profoundly impact the environment and ecosystems. Coastal development, such as harbor and fishing port reclamations, exacerbates erosion through various mechanisms. Altering the natural ecosystem and coastal topography leads to sediment
displacement by wave action and nearshore currents, resulting in coastal erosion. These marine life habitats contribute to water



pollution, significantly affecting marine ecosystems and jeopardizing coastal infrastructure, housing, and facilities, thereby increasing the risk of property loss and safety hazards (Airoldi and Beck, 2007).

Coastal erosion is currently emerging as a global problem, as several studies point out. In particular, climate change is causing sea levels to rise and coastal erosion to be a new environmental problem. Rates of coastal loss due to sediment transport are

increasing every year, which is critical for highly exposed coastal cities and coral islands vulnerable to erosion (Ortega et al., 2023; Parvathy et al., 2023). As technology develops and data accumulates, remote sensing has become a very effective means of analyzing coastal erosion. Remote sensing technology allows the collection and analysis of high-resolution images through a variety of platforms, including satellites, aircraft, and drones. This technology allows you to monitor changes in the shoreline in real time or precisely analyze past erosion levels and patterns (Nativí-Merchán et al., 2021). Several erosion mitigation

measures are also being continuously devised to counter shoreline deformation induced by coastal development or sea level rise. Beach nourishment, involving the supplementation of substantial amounts of sand to preserve the original beach, is common practice, along with the installation of coastal structures such as detached breakwaters or groins. However, implementing inappropriate coastal structures has sometimes exacerbated erosion. For instance, the double headland method was employed to counter sand loss at Yeongrang Beach, Sokcho City, Gangwon Province. Nonetheless, as noted by Lim et al.

(2021), it resulted in excessive diffraction waves, which impeded the intended erosion reduction function of the coastal structure.

Changing wave fields caused by ports and coastal structures influence coastal sediment transport, leading to shoreline alterations and erosion. Numerous studies have elucidated sediment transport in coastal areas due to wave action and established theoretical formulas for coastal erosion. Pelnard-Considère (1956) proposed a governing equation wherein the

shoreline position is determined by longshore sediment transport (LST) along the coast. This equation is based on the assumption that sediment does not alter the beach profile and that the active profile of the beach uniformly advances or retreats in the transverse direction. In reality, LST is considered more influential than episodic cross-shore sediment transport in driving significant shoreline changes over extended periods. Predictions of shoreline changes solely using empirical formulas for LST are limited by the complex causes underlying the shoreline alterations caused by ports and coastal structures. For instance,

during storm events, wave breaking suspends beach sand, leading to significant short-term erosion in the lateral direction. Yates et al. (2009) investigated shoreline equilibrium on coasts eroded by suspended sediments under constant wave energy influx using field observation data. Although long-term field observations yield insightful results, Kim et al. (2021) proposed a simple method that achieved similar outcomes, by applying an empirical formula for equilibrium beach profiles. When wave activity subsides post-storm, suspended sand settles, forming berms and restoring the original coastline. Lim et al. (2022) and

Lim and Lee (2023) derived governing equations for simulating both long-term and short-term shoreline erosion caused by LST, analyzing short-term coastal erosion with the horizontal behavior of suspended sediments.

The one-line shoreline change model introduced by Pelnard-Considère (1956) specializes in simulating temporal shoreline changes owing to groin installation despite several limitations (Walton and Chiu, 1979; Le Mehaute and Soldate, 1979; Hanson, 1989). However, the original version does not consider wave diffraction effects caused by large breakwaters or detached





breakwaters. Consequently, efforts have been made to enhance the model in scenarios where wave diffraction from coastal structures is significant. Various numerical (Hanson, 1989; Leont'yev, 1997, 2007) and mathematical (Vaidya et al., 2015) approaches, primarily based on empirical formulas, have been proposed. The GENESIS model proposed by Hanson (1989) incorporates the impact of coastal structures on shoreline changes by introducing an additional term for longshore variation of breaking wave height, as presented by Ozasa and Brampton (1980). Although this model is widely used in engineering

consulting, it underestimates results in scenarios with significant wave diffraction from reclaimed revetment or breakwaters constructed outside ports (Lee and Hsu, 2017).

Recently, Lim et al. (2021) developed a shoreline change model by applying the empirical equilibrium shoreline formula proposed by Hsu and Evans (1989) to reflect wave diffraction from capes, bays, and artificial structures. The planform of the static equilibrium shoreline exhibits a certain form due to shoreline equilibrium on a mobile sand bed, with numerous empirical

studies conducted for the prediction (Hsu and Evans, 1989; Moreno and Kraus, 1999; Yasso, 1965). The parabolic bay shape equation (PBSE) proposed by Hsu and Evans (1989) is globally adopted for coastal management owing to its efficacy (González and Medina, 2001; Herrington et al., 2007; Bowman et al., 2009; Silveira et al., 2010; Yu and Chen, 2011). Lim et al. (2019) demonstrated PBSE's applicability to the East Sea shoreline of Korea using East Sea wave data. To address PBSE's control point uncertainty defined in orthogonal coordinate systems, Lim et al. (2022a) supplemented it for application in

cylindrical coordinates.

This study investigated the complete disappearance of sand on Wolcheon Beach within a year due to LST following a considerable change in the wave field resulting from large-scale reclamation on Hosan-Wolcheon Beach near Samcheok, Korea, for constructing a liquefied natural gas (LNG) pier. The shoreline change model developed by Lim et al. (2021) was employed for the analysis, comprising three main parts. First, information on the Samcheok LNG terminal and Wolcheon

Beach, the study area, was introduced, and rapid shoreline changes on Wolcheon Beach were delineated from satellite images using the Google Earth Engine. Subsequently, the results of the numerical coastal erosion simulation model were compared with satellite analysis outcomes. Finally, the LST rate was examined through shoreline change results and compared with the empirical formula of the Coastal Engineering Research Center (CERC).

As stated, the applied numerical model is an enhanced shoreline change model based on PBSE (Lim et al., 2021). This model

was refined to incorporate the influences of diffraction waves caused by significant coastal structures. The case study underscores the importance of assessing changes in nearby shorelines before performing large-scale coastal construction projects, thus providing insights into methods that minimize potential damage. In the discussion section, the impact of groin installation for controlling coastal sediment was simulated numerically, highlighting the necessity of such experiments when predicting changes in the wave field. Consequently, this study presented an opportunity to investigate the ramifications of

harbor and fishing port development, as well as large-scale reclamation, which can alter wave fields in coastal regions, on rapid and catastrophic erosion issues.





## 2 Study site

### 2.1 Location of Wolcheon Beach

Wolcheon Beach, the study site, is located in Samcheok City, Gangwon Province, Korea. Hosan Beach where the Samcheok

LNG terminal was constructed is north of the Gagok Creek and Wolcheon Beach which mainly suffered erosion damage is
located to the south of the creek. Hosan Beach is located to the south of Hosan Creek, and it formed an almost straight 1.91
km shoreline along Wolcheon Beach.

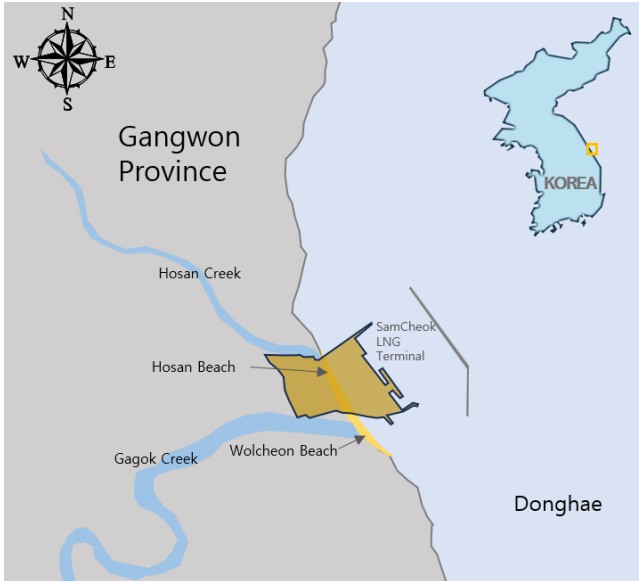

**Figure 1: Location of the study site including Hosan and Wolcheon Beaches and Samcheok LNG terminal.**

### 2.2 Overview of Samcheok LNG terminal development

The Samcheok LNG terminal is the fourth natural gas production facility in Korea after the Pyeongtaek, Incheon, and
Tongyeong terminals. It was constructed from 2010 to 2017 for the stable supply of gas in the Gangwon and Yeongnam regions.
It has a total area of approximately 980,000 $m^2$, and 590,000 $m^2$ of it is occupied by the marine site. The marine site was
completed in 2011, and 12 LNG storage tanks (three 270,000 kiloliter tanks and nine 200,000 kiloliter tanks) were installed at

the site. In addition, docking facilities for 200,000-ton LNG ships and a trade port equipped with a 1,800m breakwater, which
is the largest in Korea, were developed.

As shown in Figure 2(a), Wolcheon Beach was well preserved in a direction perpendicular to the dominant direction of wave
incidence in 2011 when public water reclamation began. In 2012, however, all of the sand on the approximately 40-m-wide
beach was lost due to the catastrophic beach erosion caused by the change in wave field as can be seen in Figure 2(b). This

became a major social issue because of the serious overtopping and erosion damage that occurred in nearby villages, prompting
the introduction of laws and systems for the assessment of the effects of beach erosion in advance when coastal area



development, such as reclamation and port construction, is planned. Lim et al. (2021) revealed that the erosion of Wolcheon Beach was caused by the reclamation project of the Samcheok LNG terminal and the outer breakwater by applying the shoreline change model, which was established by applying the PBSE of Hsu and Evans (1989) to cylindrical coordinates.

**(a)**  **(b)**

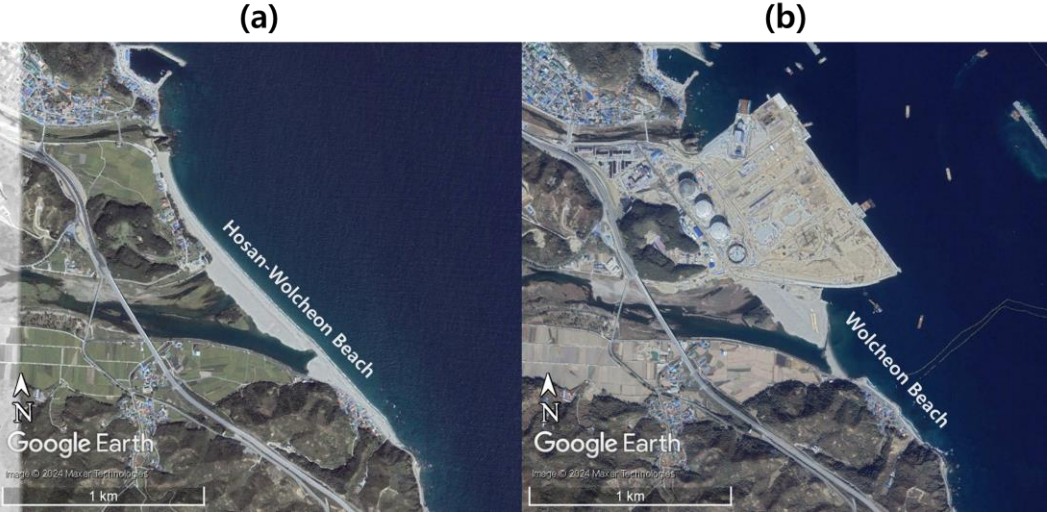


**Figure 2: Aerial photographs of Hosan-Wolcheon Beach before the construction of the Samcheok LNG terminal: (a) 2011.06; (b) 2012.10 (© Google Earth).**

**2.3 Wave and coastal environment of Wolcheon Beach**

The coastal waters of Samcheok where Wolcheon Beach are deep (maximum depth of 3,000 m or higher; average depth of
1,300 or less) and have high waves. The coast is elevated, and it is long and straight because the dominant direction of wave incidence is almost perpendicular to the shoreline. Regarding the incident wave to Wolcheon Beach, the root mean square (RMS) wave height is estimated to be 1.14 m as can be seen from the National Oceanic and Atmospheric Administration (NOAA) data in Figure 3. The NOAA data site is located at 37.0 °N, 129.5 °E, 27 km from Wolcheon Beach. Figure 4 shows the wave rose (blue) in deep water obtained from wave hindcasting data and the resulting rose diagram of LST components
(green: north, orange: south). The static equilibrium shoreline is considered to occur at the angle at which LST is balanced. The dominant direction of wave incidence for the static equilibrium of Wolcheon Beach was found to be 34.2 N from the true north. The direction of the static equilibrium shoreline maintains an angle of approximately 90 degrees with the dominant direction of wave incidence in the absence of the net transport component of LST.




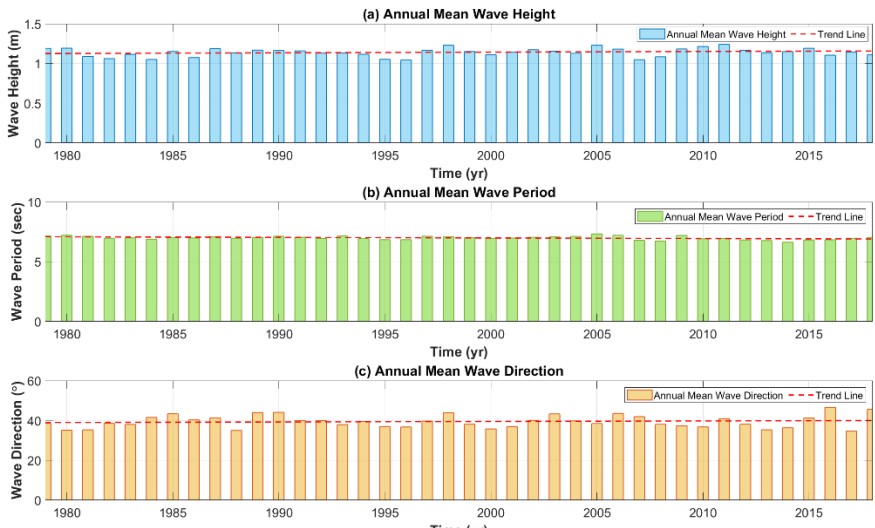

**Figure 3: Changes in the annual mean values of the wave height (a), wave period (b), and wave direction (c) of NOAA wave data near Wolcheon Beach.**

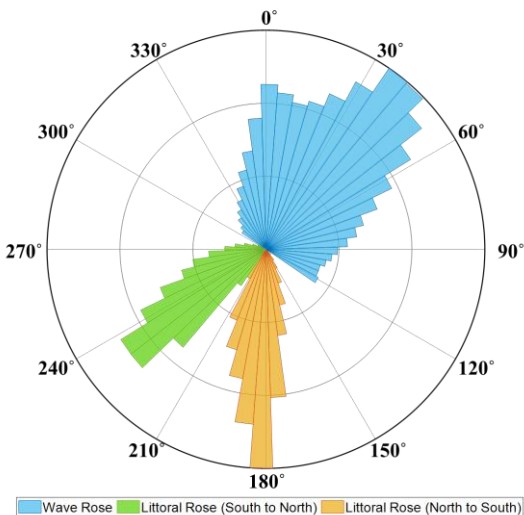

**Figure 4: Combined rose diagram of wave and littoral drift for the study site.**

In addition, the tidal range is very small (30 cm or less) in Samcheok coastal waters. The waters are significantly affected by the flow of waves rather than the tide. The mean sea level (S0), which is the sum of the four major partial tides, in the waters near the Hosan port is significantly low (18.4 cm). The tide form number, that is, the ratio of the diurnal tide semi-tidal range (K1+O1) to the semidiurnal tide semi-tidal range (M2+S2), is 1.45 cm, indicating that the semidiurnal tide is dominant and that two high tides and two low tides occur every day. The impact of the tidal current was nonsignificant as the tidal range was approximately 0.3 m. Southward flow was observed during the flood tide and northward flow during the ebb tide with 0.1 to 0.4 m/sec.





## 3. Beach survey data

### 3.1 GPS shoreline survey

Analysis of the data of the shoreline survey conducted six times from April 2010 to December 2011 with rapid shoreline changes revealed that the beach width increased to 60 m in the baseline 04 section of Hosan-Wolcheon Beach where the Gagok

Creek was located in approximately 1.5 years, as shown in Figure 5, however, it decreased by more than 90 % in the baseline 05 and 06 sections where Wolcheon Beach was located. This indicates the occurrence of a coastal erosion problem that considerably threatens the lives, properties, and livelihoods of the people living behind Wolcheon Beach.

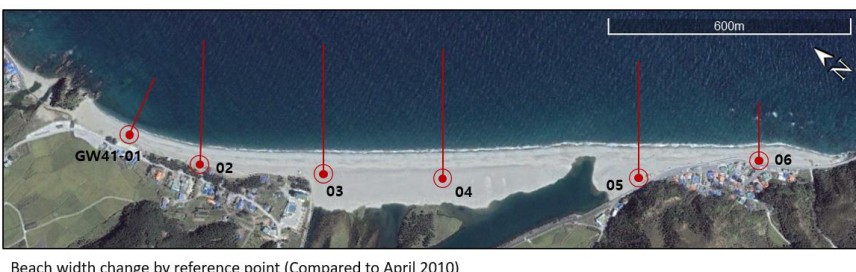

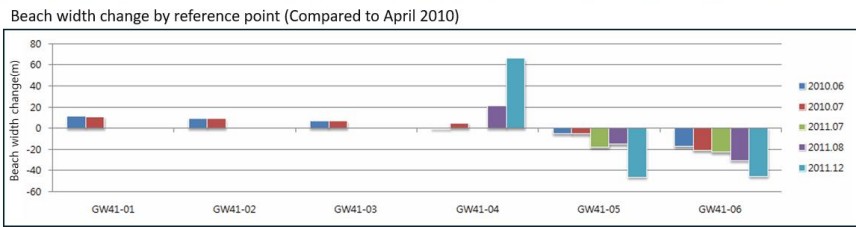

**Figure 5: Beach width change by reference point on Hosan-Wolcheon Beach (source: GSESRH, 2012; © Google Earth).**

### 3.2 Sand characteristics survey

The sand grain size on a beach is an important physical variable used to determine the LST rate, which is formed by the action of wave energy on the coast. Towing to the construction of the Samcheok LNG terminal, most of the sand on Wolcheon Beach was introduced into the estuary of the Gagok Creek in 2012 as shown in Figure 6(a). Figure 6(b) shows the cumulative sand grain size distribution surveyed during the period. The median grain size of sand ($D_{50}$) is 0.666 mm. The porosity and specific

gravity of sand are also required when determining the LST rate. Owing to the absence of the data in the target area, porosity $p = 0.42$ and sand gravity weight $s = 2.65$, which were obtained from the sand sample collected from the sandy beach located in the same watershed system were applied.



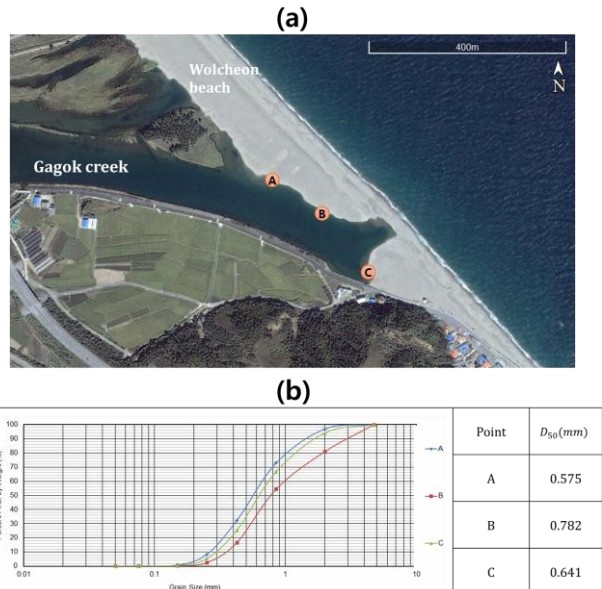

**Figure 6: Sand collection location (a) and survey results (b) for cumulative grain size curves and median grain sizes for sand ($D_{50}$)**
**at the Gagok estuary (source: GSESRH, 2013; © Google Earth).**

### 3.3 Analysis of the shoreline data acquired from satellite images

To identify rapid shoreline changes on Wolcheon Beach between 2011 and 2012, Landsat-7 images from the Google Earth engine platform were used. Landsat-7, launched in 1999, is managed by the United States Geological Survey. The Landsat series captures high-resolution multispectral images of the Earth's surface and provides important information in various areas, including environmental monitoring, natural disaster monitoring, agriculture, forest management, and urban development. They have red, blue, green, near-infrared, and medium infrared spectral bands with a 30 m spatial resolution. RGB images at ten-time points in which changes in shoreline can be observed were selected, as shown in Figure 7. Several authors (18–20) successfully used radar images for shoreline detection and extraction purposes. However, this often requires rigorous terrain correction, geocoding, and radiometric correction and balancing (21). To extract shoreline, several authors (Baghdadi et al., 2004; Modava and Akbarizadeh, 2017; She et al., 2017; Vos et al., 2019; Bengoufa et al., 2021) have successfully used satellite images. However, as Liu and Jezek (2004) have pointed out a long time ago, this often still requires rigorous terrain correction, geocoding, and radiometric correction and balancing.

However, as shown in the figure, However, as you can see in the figure, the repeated occurrence of black bands that did not allow the images to be read made it difficult to process most of the images with conventional shoreline extraction methods. Therefore, images were extracted using the direct digitizing method. The LNG terminal's revetment and the extracted shorelines are also shown on the corresponding satellite imagery, as shown in Figure 7, and the entire extracted shorelines are shown in the last satellite image in Figure 8 from the shoreline on March 13, 2011, before the LNG project began, to December




12, 2012, approximately 21 months later. It was revealed that in less than two years, an average of 37 m of beach erosion occurred in Wolcheon Beach, while a shoreline advance of 145 m occurred along the LNG terminal revetment.

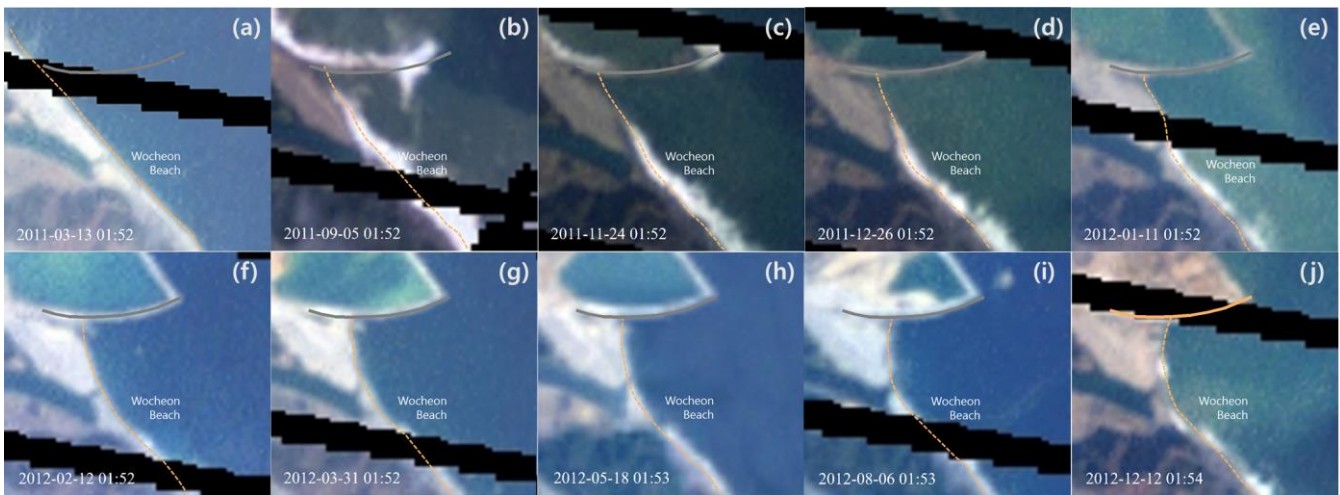

185

**Figure 7: Satellite images of Landsat-7 selected for Wolcheon Beach shoreline analysis between 2011 and 2012, and the extracted shorelines: (a) 2011.03.13; (b) 2011.09.05; (c) 2011.11.24; (d) 2011.12.26; (e) 2012.01.11; (f) 2012.02.12; (g) 2012.03.31; (h) 2012.05.18; (i) 2012.08.06; (j) 2012.12.12 (© Google Earth).**

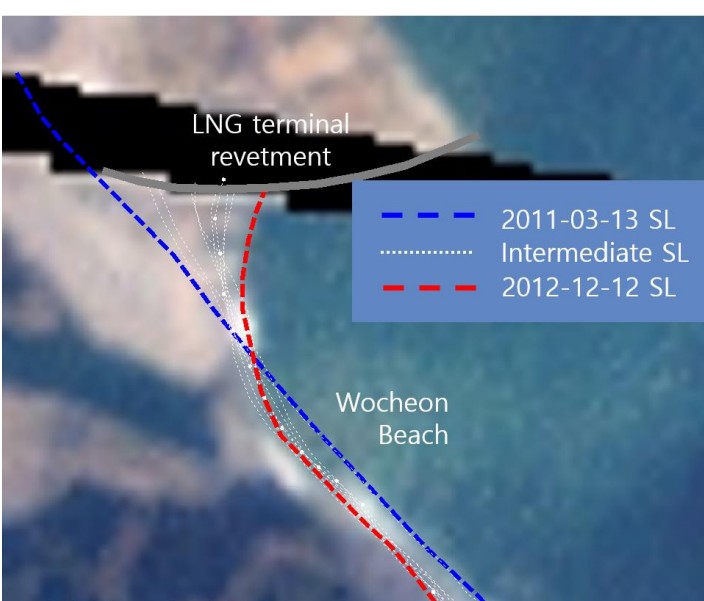

190 **Figure 8: Shoreline changes from satellite images of Landsat-7 in Wolcheon Beach between 2011 and 2012 (© Google Earth).**



## 4. Numerical simulation of shoreline change

### 4.1 Governing equation

In this study, we employ one of the currently available shoreline change models that can well simulate the temporal change of the shoreline extracted from satellite images. Recently, Lim et al. (2021) extended the governing equation first proposed by Pelnard-Considère (1956) to cylindrical coordinates as given below for an application to concave coasts, as most coasts are.

$$\frac{\partial r_s}{\partial t} + \frac{1}{(h_c+h_B)} \frac{\partial Q_{l,\theta}}{r_s \partial \theta} = 0 \tag{1}$$

where $r_s$ is the distance from the center of the circumference to the shoreline, decreasing and increasing when the shoreline advances and retreats, respectively. The $\theta$ represents the coordinates in the shoreline direction, and $h_B$ and $h_c$ are the berm height and closure depth, respectively. The $Q_{l,\theta}$ is the longshore sediment transport (LST) rate in the $\theta$ direction. To consider the wave diffraction effect caused by the presence of structures, the LST rate equation can be modified based on CERC (1984) the as follows.

$$Q_{l,\theta} = C'H_b^{\frac{5}{2}} sin\, 2(\alpha_m - \alpha_e) \tag{2}$$

where $H_b$ is the breaking wave height and $\alpha_b$ is the wave incident angle at the breaking point, $\alpha_m$ is the annual mean wave angle, and $\alpha_e$ is the equilibrium planform gradient which can be estimated based on the approximate PBSE. In Eq. (2), $C'$ is a constant calculated by

$$C' = \frac{K\sqrt{g/\kappa}}{16(s-1)(1-p)} \tag{3}$$

where $K$ is the coastal sediment coefficient, which can range between 0.04 and 1.1, depending on the sediment transport. Here, Komar and Inman (1970) proposed $K$ as a value of 0.77. And $s$ and $p$ are the sediment specific weight and the sediment porosity, respectively.

### 4.2 Estimation of initial LST by PBSE

When a structure such as a breakwater or a groin is installed on the shore, the equilibrium shoreline changes, and LST is generated towards the structure. As shown in Eq. (2), the LST has a maximum value when $(\alpha_m - \alpha_e)$ is 45° in degree unit. Since $\alpha_m$ is 0 degrees initially, the maximum LST value will initially be shown at the shoreline location where $\alpha_e$ becomes 45°. In Eq. (2), the equilibrium planform gradient $\alpha_e$ can be obtained from the approximate expression of PBSE given below (Figure 9).

$$R \cong \frac{a}{sin\,\beta} \frac{\beta}{\theta_e} \tag{4}$$




where $a$ denotes the vertical distance between the wave crest baseline passing through the focus point and the shore baseline passing through the downdrift control point X (i.e., down-coast limit), $\theta_e$ is the angle between the wave crest baseline and the line connecting the parabolic focus to the equilibrium shoreline, and $\beta$ is the reference wave angle at the downdrift control

point.

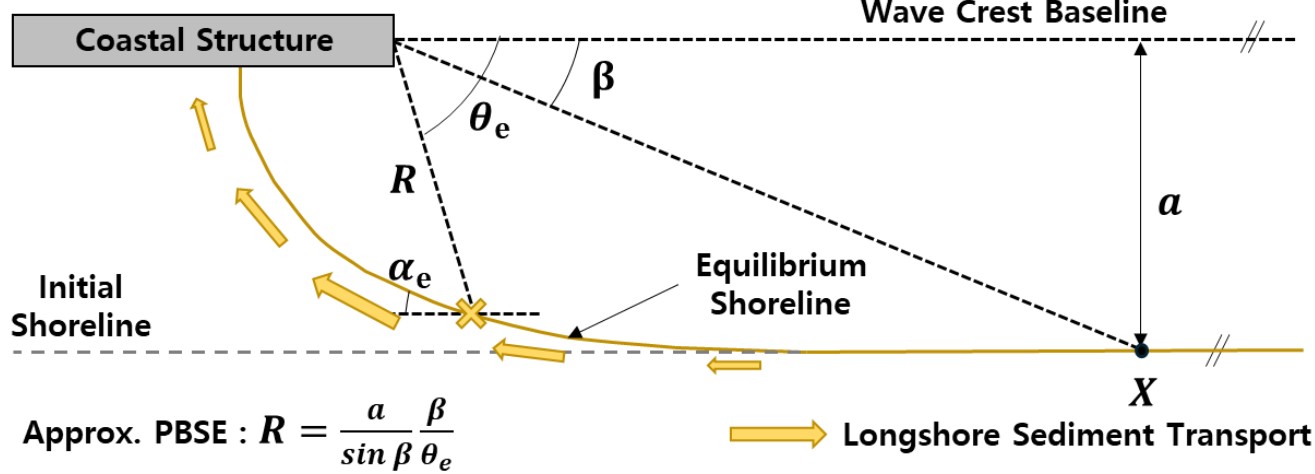

**Figure 9: Definition sketch for estimation of initial LST owing to construction of coastal structure.**

If the coast is ultimately eroded due to LST, it can be assumed that the $\theta_e$ at which erosion occurs on the coast becomes $\beta$. Therefore, $\theta_e$ can be obtained from the straight distance, $R$, between the focus and the grid point of the $\theta$ cell, as shown in the

equation below.

$$\theta_e \cong tan^{-1}\left(\frac{a}{R}\right) \tag{5}$$

The $\alpha_e$ can be estimated based on the approximate PBSE according to $\theta_e$, using the following equation.

$$\alpha_e = tan^{-1}\left(\frac{sin\,\theta_e - \theta_e\,cos\,\theta_e}{cos\,\theta_e + \theta_e\,sin\,\theta_e}\right) \tag{6}$$

Figure 10 shows the dimensionless initial LST ($Q_{l,\theta}/(C'H_b^{5/2})$) according to $\alpha_e$ obtained by applying Eq. (6) to the CERC

equation after the installation of coastal structures. In the case of Wolcheon Beach where $\theta_e$ is from 81.9 ° to 92.8 °, the dimensionless initial LST is from 0.807 to 0.933, indicating that the installation of the Samcheok LNG terminal may cause serious erosion due to LST in the Wolcheon Beach area. In particular, in the Gagok Estuary where $\theta_e$ ranges from 92.8 ° to 104.5 °, the dimensionless initial LST is calculated from 0.933 to 0.998, making it an area where the shoreline deformation due to serious LST differences is unavoidable.



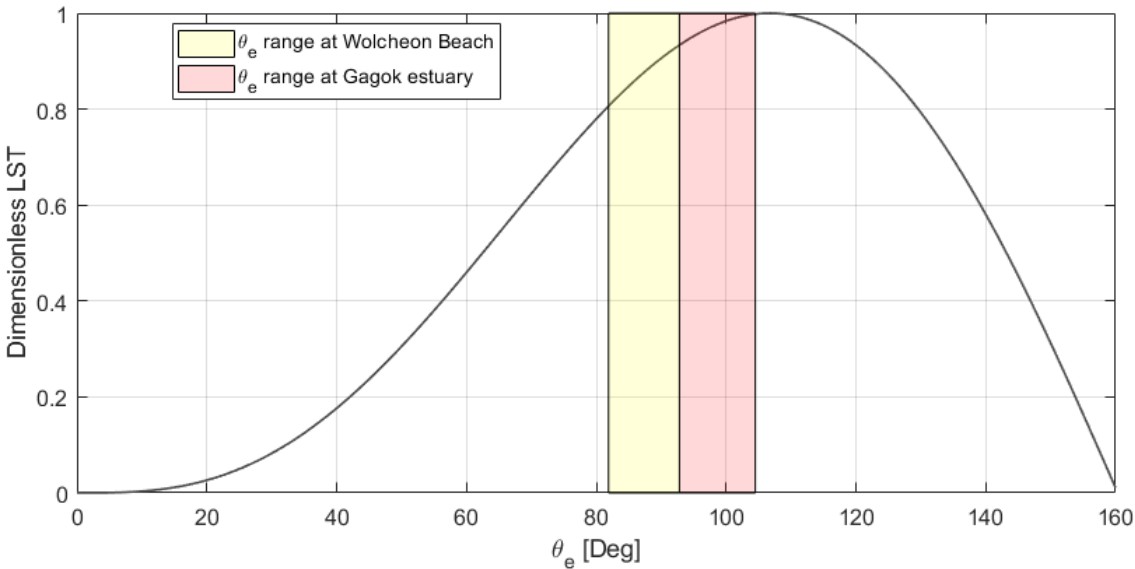


**Figure 10: Dimensionless initial LST ($Q_{l,\theta}/(C'H_b^{5/2})$) according to the shoreline location angle $\theta_e$.**

## 4.3 Numerical scheme

The governing equation of the shoreline change model was solved by the finite difference method. The beach is divided into $\Delta\theta$ grids along the coast, and it is assumed that sediment transport in the zone increases or decreases depending on the loss or

inflow of sediment by grid along the coast. A staggered grid system is used, in which $\{r_s\}$ and $\{Q_{l,\theta}\}$ are defined alternatively in odd-even order ($i$ represents the grid number). $Q_{l,\theta}$, the sediment transport along the longshore grid, was defined to be located at the boundary of each grid, while the shoreline position was defined to be located at the center of the grid. To conveniently express the finite difference equation, the superscript $n+1$ denoted the value to be obtained at the next time step, and $n$ was defined as the value already calculated at the present time step. Therefore, $r_{si}^{n+1}$, which is the shoreline position

of the $i$-th grid at the next time step $n+1$, can be expressed as Eq. (7).

$$r_{si}^{n+1} = r_{si}^n + \frac{\Delta t}{h_{i,j}}\left(\frac{Q_{l,\theta i+1}-Q_{l,\theta i}}{r_{si}\Delta\theta}\right) \tag{7}$$

where $\Delta t$ is the time step and $\Delta\theta$ is the shoreline grid. The LST rate that converges to equilibrium can be calculated using Eq. (7). As previously explained, the explicit scheme method is used to obtain the newly determined shoreline position using the past value.

The erosion control line is the boundary condition on the shoreside in the transverse direction. When the shoreline met the hard boundary, that is, the erosion control line, due to the progress of erosion, the complete loss of the beach sand was assumed for no further generation of longshore sediment and no more retreat of the shoreline.




## 5. Application to the catastrophic beach loss event on Wolcheon Beach

### 5.1 Review of changes in equilibrium shoreline after Samcheok LNG reclamation

The Samcheok LNG terminal near the study site was constructed after large-scale reclamation. Such reclamation changes the equilibrium shoreline, as shown in Figure 11. The equilibrium shoreline was estimated using MeePaSoL of Lim et al. (2022), a Matlab GUI tool. After reclamation, the equilibrium shoreline is changed as marked in green due to the change in wave environment. It corresponds to the shoreline to be newly formed on average if it is composed of sand. Therefore, the sand on the sea side of this line is subjected to LST towards the estuary of the Gagok Creek due to diffraction waves.

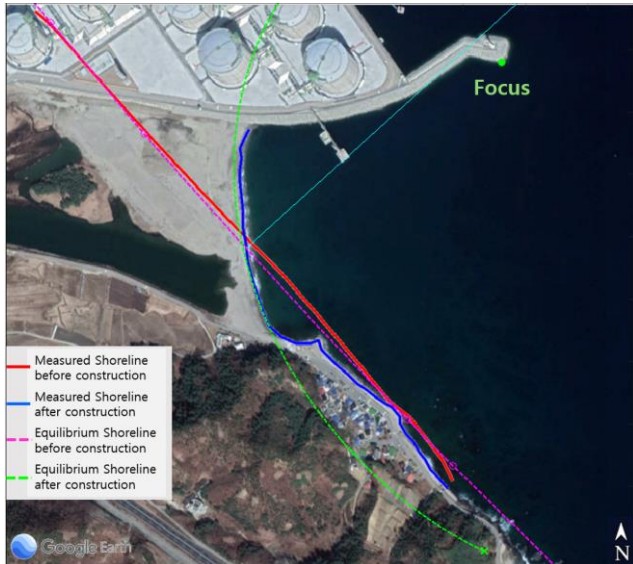


**Figure 11: Change in equilibrium shoreline after reclamation estimated using MeePaSoL (© Google Earth).**

### 5.2 Numerical simulation conditions

According to the information obtained through the shoreline analysis in Subsection 3.3, the shoreline began changing from March to September in 2011. Through several simulations, the most similar start time for the shoreline change was found in

the numerical model results and the satellite data analysis results. Based on this, it was inferred that shoreline deformation began on August 17, 2011. Table 1 shows the values used in the numerical simulation. Numerical simulation was performed for the wave height $H = 1\ m$ and wave period $T = 5\ sec$ as normal wave conditions. This is because the shoreline change due to the wave diffraction effect caused by coastal reclamation or the installation of port or coastal structures lasts for long periods due to the LST by the oblique inflow of waves under ordinary wave conditions with a low wave height rather than the

deformation by high waves.

Figure 12 shows the area and grid information applied to the numerical model. The coordinates of the origin of the cylindrical coordinate system are 37°11'49'' N and 129°23'45'' E, and the radius for fitting the shoreline of Hosan-Wocheon Beach is $R = 5.270\ km$. The computing area was composed of 50 grids at $\Delta\theta = 0.2176°$ intervals along the 1.0-km-long beach zone




from $\theta_s = 81.9°$ to $\theta_e = 92.8°$ with respect to the true north for the center of the circle that fitted the original shoreline. At the

points where the shoreline is located, $\Delta\theta$ corresponds to a length of 20 m. The southeastern boundary had a boundary condition that the inflow and outflow of sand are free, and an impermeable boundary condition was applied along the LNG revetment.

**Table 1: Values applied in numerical simulation.**

| Inputs | LST constant $C'$ | Breaking wave height $H_b$ | Number of grids $n$ | Radius fitting the original shoreline $R$ |
|---|---|---|---|---|
| Values | 0.178 | $1\ m$ | 50 | 5.27 km |

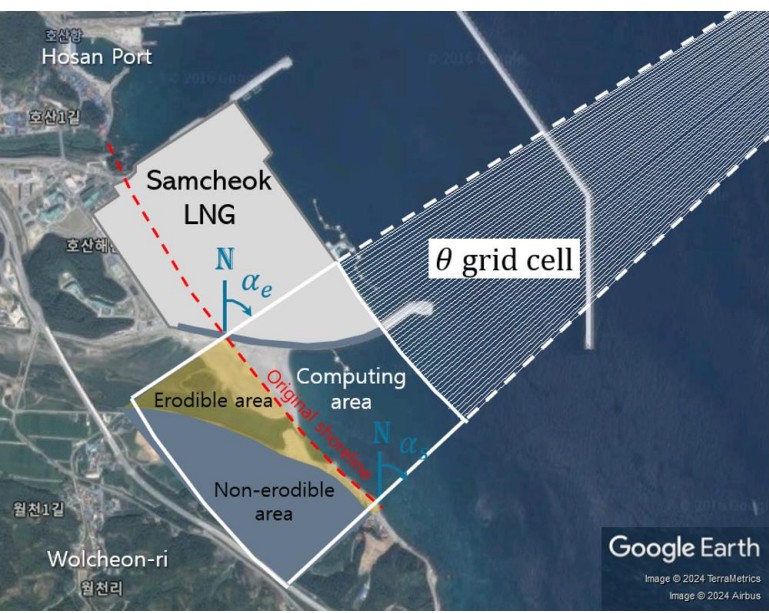

**Figure 12: Grid system of cylindrical coordinates for shoreline change numerical simulation (© Google Earth).**

**5.3 Numerical simulation results and verification**

**5.3.1 Shoreline change prediction results**

It was assumed that the shoreline change began from August 2011 when the revetment for Samcheok LNG terminal reclamation was completed, and numerical simulation was performed until December 2012 when further significant changes

were not observed. A value of approximately 0.1782 is generated for $C'$ under the application of the coastal sediment coefficient $K$ =0.77, wave breaking coefficient $\kappa$ =0.78, sediment specific weight $s$ =2.57, and porosity $p$ =0.39, which are applied to the typical sediment transport rate. On Koreas east coast, the specific gravity $s$ =2.65 and porosity $p$ =0.42 were obtained on average. Therefore, $C'$ =0.1783, which is the commonly used value, was considered.

The numerical model results were displayed in yellow dash lines on satellite images as shown in Figure 13. The gray line

represents the revetment. In the numerical simulation, when the shoreline reached the revetment, the shoreline was set to have no more change. Overall, the shoreline recognized from satellite images was similar to the numerical simulation results. Since


the grid size of the numerical model was 20 m, the satellite data were interpolated at the same interval. Figure 14 shows the shoreline changes extracted from satellite images from March 3, 2011 to December 12, 2012, on the satellite images of December 12, 2012. An average of 22 m of beach erosion occurred in the erosion area of Wolcheon Beach, and a maximum

shoreline advance of 102 m occurred at the revetment of the LNG terminal. The numerical results are underestimated compared to the values obtained from satellite images. Figure 15 compared the numerical calculations with the values obtained from satellite images.

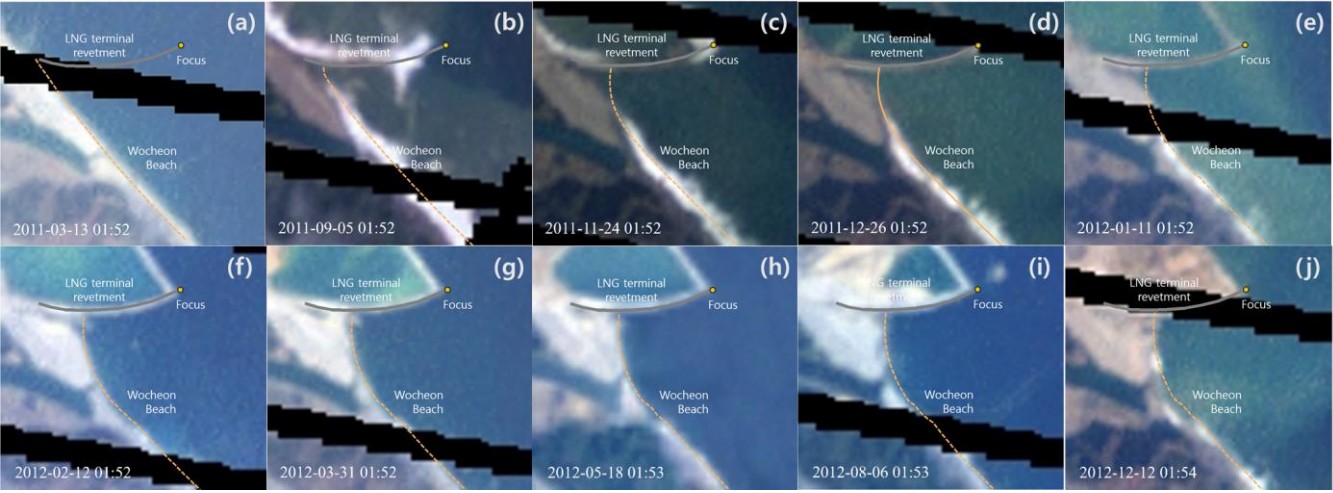

**Figure 13: Comparison between the predicted shoreline change model results and the shoreline extracted from satellite images: (a)**
**2011.03.13; (b) 2011.09.05; (c) 2011.11.24; (d) 2011.12.26; (e) 2012.01.11; (f) 2012.02.12; (g) 2012.03.31; (h) 2012.05.18; (i) 2012.08.06;**
**(j) 2012.12.12 (© Google Earth).**

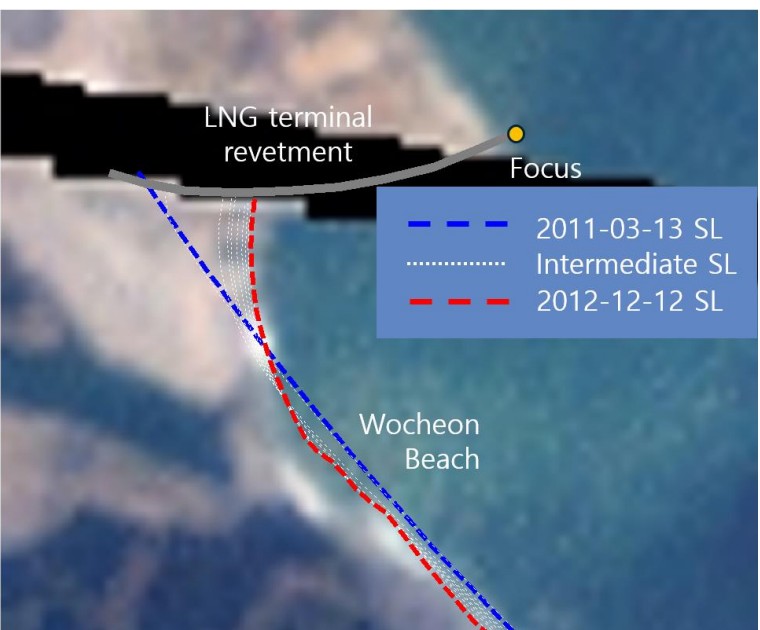

**Figure 14: Numerical results of shoreline changes in Wolcheon Beach from Mar. 13, 2011 to Dec. 12, 2012 (© Google Earth).**




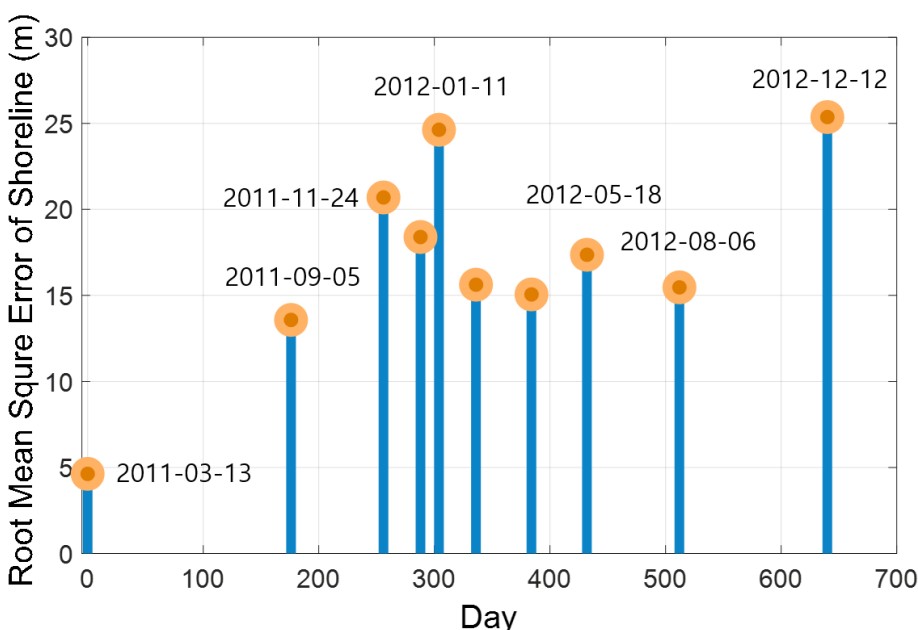

**Figure 15: Temporal variation of root mean square errors between the shoreline obtained from satellite images and the predicted shoreline results.**

**5.3.2 Comparison through LST rate vectors**

This section presents the results of the generation of the LST rate from the shoreline change extracted from satellite images. Similar research has been conducted by Jung et al. (2004) and (Rahmawati et al., 2021). The results of the equation below, obtained from the finite difference equation of the shoreline change model, are obtained from the shoreline data at two adjacent time points. In this study, the shoreline change by grid was estimated from coastal observation data by applying the formula proposed by Jung et al. (2004) to the shoreline data, which were modified to maintain the beach area based on the sediment mass conservation rule.

Jung et al. (2004) presented a method of estimating the LST rate $Q_{l,i}$ under the given shoreline change width $\Delta y_i$ as follows. Accordingly, the LST rate during the shoreline survey period $\Delta t$ is calculated.

$$Q_{l,i+1} - Q_{l,i} = \Delta x D_s \frac{\Delta y_i}{\Delta t} = C \Delta y_i = \acute{C}_i \tag{8}$$

where $C = \Delta x D_s / \Delta t$ and $D_s = h_c + h_B$. Thus, if $C_i' = \Delta x D_s \Delta y_i / \Delta t$ holds and the upstream LST rate $Q_1$ is known, a matrix equation to estimate the LST rate $Q_{l,i}$ is derived as follows.

$$\begin{bmatrix} 1 & & & \\ -1 & 1 & & \\ & : & & \\ & & -1 & 1 & \\ & & & -1 & 1 \end{bmatrix} \begin{bmatrix} Q_{l,2} \\ Q_{l,3} \\ : \\ Q_{l,N} \\ Q_{l,N+1} \end{bmatrix} = \begin{bmatrix} C_2' + Q_{l,1} \\ C_2' \\ : \\ C_{N-1}' \\ C_N' \end{bmatrix} \tag{9}$$





The validity of the model results was examined by comparing the vectors obtained from the results of the numerical model. Figure 16 compares the LST rate vectors obtained from two temporally adjacent aerial photographs with those obtained from the model. A considerable amount of the sand on Wolcheon Beach is headed towards the estuary of the Gagok Creek due to the reclamation project for the Samcheok LNG terminal. For each figure, the reference vectors have been adjusted accordingly so that the vector patterns can be compared to each other. The numerical model results show a consistent LST vector pattern

towards the LNG terminal revetment except for the initial results, but the results obtained from satellite images do not always show a vector towards the LNG terminal revetment due to the effect of transient high wave inflow.

Figure 17 compares LST vectors with each other on average over the entire analysis period (from Mar. 13, 2012 to Aug. 6, 2012). The results of the analysis from the satellite images show that the direction of the vector changes in the short term due to the influence of high waves, but the results are almost the same, so it can be seen that the numerical model results reproduce

the phenomenon of LST generation towards LNG revetment well due to the LNG terminal reclamation project.


**Figure 16: Comparison between the LST vectors obtained from satellite images (left; red boxes) and the numerically simulated LST vectors (right; blue boxes): (a) 2011.03.13 – 2011.09.05; (b) 2011.09.05 – 2011.11.24; (c) 2011.11.24 – 2011.12.26; (d) 2011.12.26 – 2012.01.11; € 2012.01.11 – 2012.02.12; (f) 2012.02.12 – 2012.03.31; (g) 2012.03.31 – 2012.05.18; (h) 2012.05.18 – 0.12.08.06.**





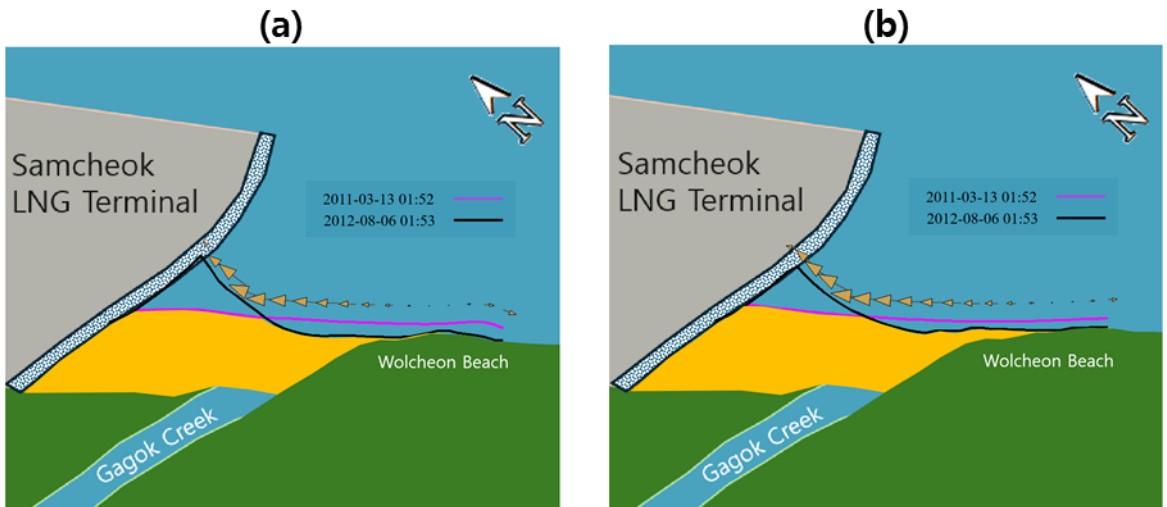

**Figure 17: Comparison of LST vector results over the entire analysis period: (a) results from satellite images; (b) Results from shoreline change model.**

Figure 18 compares the averaged magnitude of the LST vectors for each grid and the cumulative amount of longshore sediment towards the LNG revetment. The positive number represents LST towards the LNG revetment, while the negative number

indicates LST towards the opposite direction to the LNG revetment. The results of the satellite images showed severe undulation compared to the numerically simulated results, so the results obtained by smoothing through the front-and-back values as expressed in the following equation were shown together with the dotted line. As a result, it can be seen that there is a fairly similar trend compared to the numerically simulated results.

$$\bar{Q}_{ls,n} = \frac{(\bar{Q}_{l,n+1} + \bar{Q}_{l,n} + \bar{Q}_{l,n-1})}{3} \qquad (10)$$

where $Q_{ls}$ is the smoothed value of averaged LST and the subscripts, $n+1$ and $n-1$ imply the value immediately following and before the $\bar{Q}_l$ value of the $n^{th}$ time.





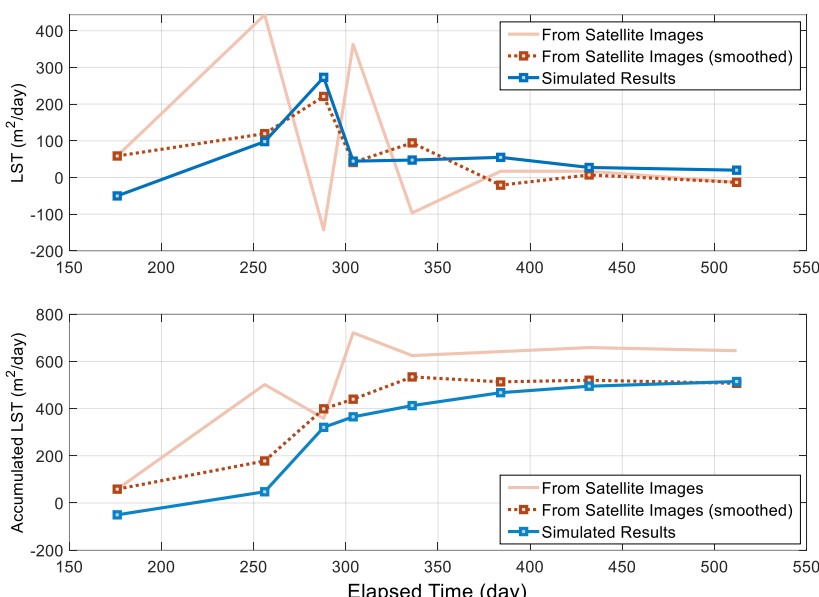

**Figure 18: Comparison between the magnitudes of LST obtained from satellite images and LST numerically simulated (upper figure), and cumulative amount of longshore sediment towards the LNG revetment (lower figure).**

## 6. Discussion

When the reclamation project was planned, action was not taken owing to the absence of means with which to predict such large-scale erosion in advance. Therefore, this section discusses appropriate measures that can be taken after assessing the impact of the construction of LNG revetments on the rotation of the shoreline and the scale of LSTs by applying PBSE, which predicts equilibrium shorelines. If LST occurred due to the change in wave field, groins can serve as representative coastal structures for LST control. Therefore, the effect of groin installation before performing the LNG project on preventing the sand loss on Wolcheon Beach was examined. In addition, since the large shoreline rotation occurs as identified in the equilibrium shoreline prediction and the installation of a single groin cannot achieve satisfactory performance, the effect of installing a groin group was also examined. The construction of the LNG revetment caused shoreline rotation at each point of Wolcheon Beach. To obtain the rotation angle ($\alpha_e$) that ultimately converges, Eq. (4) (Subsection 4.2) was applied.

In the PBSE equation, it was assumed that the focus was located at the end of the LNG revetment shown in Figure 11 and that the control point was located on the original shoreline of each $\theta$ grid cell. Rotation angle $\alpha_e$ provides the information required to approximately calculate the protrusion length of the groin to prevent sand loss from the beach with a beach width of $W$ and a length of $L_B$, as shown in Figure 19. In performing this calculation, the shoreline position in the groin is slightly inner compared to the seaward end point of the groin, but it was assumed to be located at the groin end, as shown in Figure 20.




Figure 20 also shows the rotation angle $\alpha_e$ obtained from the Wolcheon Beach area located to the southeast of the Gagok estuary under these conditions and the groin interval calculated accordingly.

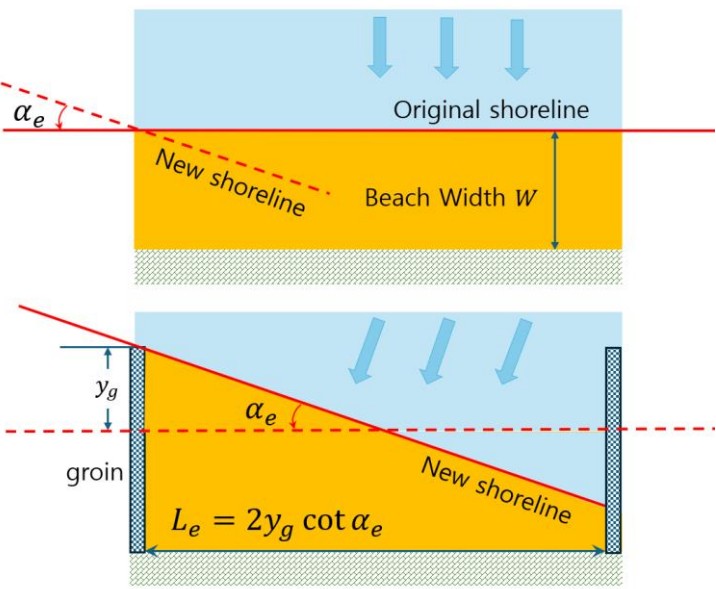

**Figure 19: Rotation of equilibrium shoreline (upper figure) and beach preservation concept by groin installation (lower figure).**

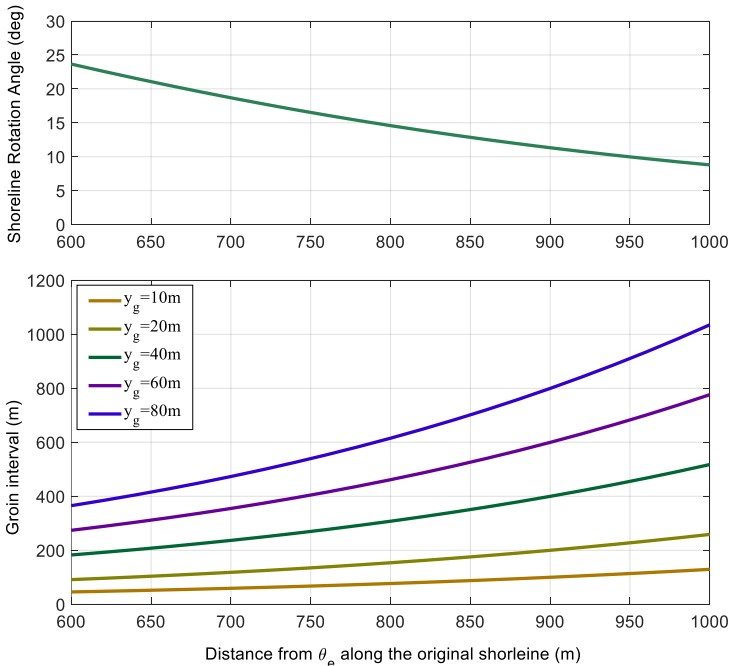

**Figure 20: Shoreline rotation angle results according to the θ grid cell (upper figure) and the groin interval to prevent the sand loss by LST (lower figure).**




The original sand cannot be maintained without protrusion length because groins are not installed. However, if groins are installed, it can be seen that the groin interval increases with their protrusion length. If it is intended to prevent the sand loss due to LST by installing a single groin on 400-m-long Wolcheon Beach, this can be achieved by installing the groin with a

value of $y_g = 70\ m$, which shows a groin interval of approximately 400 m at $x = 700\ m$, at $x = 600\ m$. If two groins with the same protrusion length are installed, $y_g = 30\ m$ that shows a first groin interval of 150 m and a second groin interval of approximately 250 m may be available. Therefore, the groin can be installed at $x = 600\ m$ and $x = 750\ m$. Likewise, if three groins are installed, $y_g = 20\ m$ that shows a first groin interval of 75 m, a second groin interval of 125 m, and a third groin interval of 200 m may be available. In this case, three groins can be installed at $x = 600\ m$, $x = 675\ m$, and $x = 800\ m$,

respectively. The above results show that a single groin is highly likely to cause problems due to the excessively large protrusion length for the Samcheok LNG terminal. Two groins can be acceptable, but it seems desirable to install three groins with a protrusion length of $y_g = 20\ m$. The above calculations do not guarantee sand retention on the coast that exceeds $x = 1000\ m$, which is considered outside the Wolcheon Beach area.

## 7. Conclusions

In this study, a shoreline change model was applied to the complete loss of sand on Wolcheon Beach due to the strong LST caused by a reclamation project for the construction of the nearby Samcheok LNG terminal in Gangwon Province. Regarding the numerical model applied in this study, the model presented by Lim et al. (2021) was applied. This model can reflect the diffraction waves caused by coastal structures by applying the PBSE of Hsu and Evans (1989) unlike the conventional shoreline change model (Pelnard-Considère, 1957; Hanson, 1989).

The results of the model were verified through the shoreline extracted from satellite images. Both the LST rate results obtained from satellite images and those obtained from the model confirmed that the sand on Wolcheon Beach moved to the estuary of the Gagok Creek on the largest scale during the winter season between 2011 and 2012. As such a result of comparison with satellite images, the numerical model results reproduce the phenomenon of LST generation towards LNG revetment well due to the LNG terminal reclamation project.

When the reclamation project at the Samcheok LNG terminal was planned about 10 years ago, there was no adequate means to predict such large-scale erosion in advance. Therefore, if numerical predictions like this study are carried out, various countermeasures are possible. Among them, the generation of LST due to large-scale reclamation is the main cause of erosion, so installing groin in advance can most effectively reduce erosion. Applying PBSE, a well-known formula for predicting the rotation of the equilibrium shoreline due to changes in the wave field, the effects of groin protrusion length and installation

spacing on LST control and consequent sand conservation was investigated.

The results of this study show that if a numerical model that predicts the shoreline change of parabolic bay shape by approximately including wave diffraction effects had been incorporated into the decision-making process for coastal disasters



prior to large-scale construction in coastal areas, large-scale erosion problems such as the case of Wolcheon Beach would not have occurred.

## 405 Data availability

Not applicable.

## Author contributions

Supervision, J.L.L.; Writing—original draft, C.L., T.M.L., J.L.L; Writing—review & editing, C.L., J.L.L; Data acquisition, C.L., T.M.L. All authors have read and agreed to the published version of the manuscript.

## 410 Competing interests

The authors declare no conflicts of interest.

## Acknowledgements

This research was supported by the Korea Institute of Marine Science & Technology Promotion (KIMST), funded by the Ministry of Oceans and Fisheries, Korea (RS-2023-00256687).

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
