# Peer review of "Severe beach erosion induced by shoreline deformation after a largescale reclamation project for Samcheok LNG terminal in Korea"

_Natural Hazards and Earth System Sciences, 2024_

## Referee Comment (RC1)

**Reviewer X Report on "nhess-2024-176"**

**A. General Comments**

Reviewer X has read the manuscript (MS) for "nhess-2024-176" (PDF).

This MS reports the analysis of 10 satellite images of beach changes (2021.03.13 to 2022.12.12) at Samcheok LNG terminal, verification of beach changes using numerical calculations (with longshore transport, LST) and empirical bay shape model (PBSE – Hsu and Evans, 1989 with software MeePaSoL – Lim et al., 2021), and suggestion applying groins to protect/prevent beach from erosion prior to the construction of a large-scale coastal development project.

Overall, the layout of the presentation (e.g., sections and sub-sections) is in good order and all results are valuable for coastal managers, planners, and engineering consultants on a large-scale coastal project, from which beach erosion and shoreline rotation could occur, during or after the construction, arising from changes of nearshore wave field.

The title of the MS "Catastrophic beach erosion induced by littoral drift on nearly beach after Samcheok LNG's massive coastal reclamation project" is catchy and attractive, especially in using the words "catastrophic" and "massive". In reality, beach erosion downdrift of a harbor breakwater is the norm, which has been known for decades and also documented (e.g., Hsu et al., 1993; Hsu et al., 2000; Uda, 2010). Therefore, the state of beach erosion at the scale of about 40 m may be referred as "severe", which can be expected, instead of "catastrophic".

In addition, the beach that suffered erosion is not at "nearby", but more specifically at "downdrift", or "immediately downdrift". The main cause to erosion at Samcheok was not directly "induced by littoral drift", instead, it was associated with "wave-induced nearshore circulation that transported sediment within the shadow zone of the diffracted waves in the lee of a harbor breakwater or detached breakwater" (e.g., Gourlay, 1974; 1981). In addition, the preventive strategy using groins to control erosion can be found in Hsu at al. (2000), who reported examples of Japanese experience in the 1970-80s.

**B. Specific Comments**

(L1-2) Title of the paper: Suggestion --- May be modified as

> "Severe beach erosion induced by shoreline rotation after a large-scale reclamation project for Samcheok LNG terminal in Korea"?

(L30-31) "These marine life habitats contribute to water pollution, significantly affecting marine ecosystems and jeopardizing coastal infrastructures, housing, and facilities, …": Is this statement correct?

(L43-45) "… the double headland method…, it resulted in excessive diffracted waves…": What is the "excessive" in diffracted waves?

(L47-48) "Changing wave fields caused by ports and coastal structures influence coastal sediment transport, leading to shoreline alterations and erosion.":
What was the mechanism behind this phenomenon?

(L52-53) "… LST is considered more influential than episodic cross-shore sediment transport in driving significant shoreline change over extended periods.":
Should we say that "LST" was the consequence of the change in nearshore wave field, but not the direct or primary cause to erosion at Samcheok?
(See the last paragraph under A. Specific Comments)

(L70-71) " … significant wave diffraction from reclaimed revetment or breakwaters constructed outside ports.":
Why "significant wave diffraction"? What is the "reclaimed revetment" and "breakwaters

constructed outside ports"?

(L124-127) "…The coastal waters of Samcheok…have high waves. … Wolcheon Beach, the root mean square (RMS) wave height is estimated to be 1.14 m…":
Should we classify wave height in the order of 1.14 m as "high wave"?
(See also Fig. 3a for wave height)

(L128-137) "Figure 4…the resulting rose diagram of LST components (green: north, orange: south). … The dominant direction of wave incidence for the static equilibrium of Wolcheon Beach was found to be 34.2N…" and Fig. 4:

Given the dominate wave direction was 34.2N, then the average straight shoreline in equilibrium at Wolcheon Beach before 2011.03.13 had inclination from 124.2N to 304.2N (Left panel of Fig. 2 in the MS):

(calculation: N34.2º + 90º = N124.2º; N34.2º + 270º = N304.2º).

Consequently, the LST direction in summer (South to North) should be within 90 – 124N and in winter (North to South) within 304 – 338N, respectively.

However, the directions indicated above differ from the rose diagram for LST shown in Fig. 4 in the MS, which was centered around 230N in summer (Green, South to North) and centered around 180N in winter (Orange, North to South), respectively, all showing seaward LST.

PLEASE CONFIRM/VERIFY!

(L135) Figure 3: "Annual" values for mean wave height, wave period, and wave direction" are meaningless, because they do not represent the seasonal variations (summer and winter), especially during the period of investigation between 2011.03.1 and 2012.12.12.

(L218-222, Figure 9) "…the downdrift control point $X$…":
Should point $X$ be behind the initial shoreline in Fig. 4 after beach erosion?

(L259) "…, subjected to LST towards the estuary of the Gagok Creek due to diffraction waves.":
Yes, but more explicitly, due to nearshore current circulation induced by diffracted waves, which carry sediment …..?

(L265-266) "…shoreline deformation began on August 17, 2021…": This occurred as LST moving within 90 – 112N or 90 – 124N ? (See also (L128 –137) above)

(L267-270) "This is because shoreline change…lasts for long periods due to the LST by the oblique inflow of waves under ordinary wave conditions with a low wave height rather than the deformation by high waves.":

Wordy sentence. Please revise!!

(L321-326 and Figure 16) "Figure 16 comparies the LST rate vectors… The numerical results show a consistent LST vector pattern towards the LNG terminal revetment except for the initial results, but the results obtained from satellite images do not always show a vector towards the LNG terminal revetment due to the effect of transient high wave inflow.":

Was this true (effect of transient high wave inflow)? where the high waves came?

In winter, the entire Wolcheon Beach might be covered within the shadow zone induced by the diffracted waves (please see the Samcheok LNG 20211023.jpg below; or Fig. 2b in MS), whereas in summer, oblique wave reflection from or along part of the slightly curved revetment wall could form a short-crested wave system that transported sediment out and

away from the revetment toward the beach?

[Figure]

(L327-330) "Figure 17 compares LST vectors with each other on average over the entire analysis period (from March 13, 2012 to Aug. 6, 2012)… ":
The term "on average" can only compare the end results, not the transient temporal process of this investigation during the 1.5 years, thus invalidating the purpose of comparation! Therefore, Figure 17 does not support the comparation it intended to serve.
In addition, the "entire" period was from "2021", not "2022"! A typo?

(L338-343; Figure 18 on L347-349) "Figure 18 compares the average magnitude of the LST vectors from each end grid and the cumulative amount of longshore sediment towards the LNG revetment. The positive number… towards the LNG…negative number towards the opposite direction of the LNG revetment. … showed severe undulation… by smoothing through the back-and-front values… a fairly similar trend…":

(From Figure 18) Can authors explain the reason why LST obtained from satellite images decreased during the period of 255th-290th days and 305th-330th days, respectively, and LST away (negative numbers) from the revetment during 280th-295th days and 335th-370th days, respectively?

(L351-355) "When the reclamation project was planned, action was not taken owing to the absence of means with which to predict such large-scale erosion in advance. …assessing the impact of the construction… the rotation of the shoreline… by applying PBSE… groins can serve as representative coastal structures for LST control.":

Yes, the statement is correct. Additional notes of "shoreline reshaping" or "rotation of shoreline" can be found in Klein et al. (2023).

Groin system had been installed on Oarai Harbor and Iwafune Harbor in Japan to control erosion in the 1980s (see Hsu et al., 1993; 2000).

At moderate length (protrusion), groins may be effective to mitigate local erosion, but cannot modify the curvature of an embayed shoreline given by the PBSE.

(L395-398, Conclusions) "… When the reclamation project at the Samcheok LNG terminal was planned about 10 years ago there was no adequate means to predict such large-scale erosion in advance. Therefore, if numerical predictions like this study are carried out, various countermeasures are possible. … so installing groin in advance can most effectively reduce erosion. Applying PBSE…":    Use "were"?

Agreed!

Reviewer's comment:

Since the early 1990s, there are abundant of helpful knowledge available for shoreline changes at downdrift of harbors in Japan (see Uda, 2010), but perhaps none of the members on a planning committee (from government agency, academic, engineering consults, and NGO body) in any country outside Japan have the up-to-date knowledge to deal with this type of beach erosion problem (with shoreline rotation from straight to embayed), so leading to do-nothing or construction of conventional hard structures.

Without a preventive strategy proposed at the planning stage, such as the one outlined in the MS, a large sum of taxpayer's funds will be wasted on managing the beach erosion downdrift of a harbor following a large coastal project.

Ironically, similar scenario of beach erosion has occurred in many countries, rich or poor, developed or under-developed.

**C. Technical Corrections**

Language editing is strongly recommended for the entire MS to improve the readability and the overall quality of the paper, especially on the use of "**determiner (a/an/the/this etc.**) and revise the **wordy sentences** throughout the entire MS.

(L100) "… Wolcheon Beach which mainly suffered erosion damage…": Wordy sentence?

(L111) "… a trade port equipped with a 1,800 m breakwater,": Why 'equipped'?

(L113) " … when public water reclamation began.": What is this?

(L124-125) "The coastal waters of Samcheok…are deep (maximum depth of 3,000 m or higher; average depth of 1,300 or less)":

Are the numbers (depth) correct? Use word "more"? Please note that the edge of the continental shelf has an average depth about 150 – 200 m!

(L139-140) " … in Samcheok coastal waters. The waters are significantly affected by the flow of waves rather than the tide.": Why "significantly" and "the flow of waves"?

(150-151) "the occurrence of a coastal erosion problem that considerably threatens the lives, properties, and livelihoods….": Wordy sentence!

(L157) "Towing to the construction…": Typo error?

(L160-162) "Owing to the absence of the data… gravity $s$ = 2.65, which were obtained from the sand sample collected from the sandy beach located in the sane watershed system were applied.":

Wordy sentence!

(L176) "… as Liu and Jezek (2004) pointed out a long time ago, this often still requires rigorous….": Poor sentence!

(L178) "However, as shown in the figure, However, as you can see in the figure, …":

Careless and rough without checking!

(L321) "…two temporally adjacent aerial photographs…": Use "consecutive"?

(L396-397) "…if numerical predictions like this study are[1] carried out, various[2] countermeasures are[3] possible.":

[1]Use "were"?  [2]Use "proper" or "effective"?  [3]Use "could be"?

**D. Additional References other than those already included in the MS**

Gourlay, M.R., (1974). Wave set-up and wave generated currents in the lee of a breakwater or headlands. Proc. 14th Inter. Conf. Coastal Eng., pp. 1976–1987.

Gourlay, M.R., (1981). Beach processes in the vicinity of offshore breakwaters. Proc. 5th Aust. Conf. Coastal Eng., pp.129–134.

Hsu, J.R.C., Uda, T., Silvester, R., (1993). Beaches downcoast of harbours in bays. Coastal Eng., 19, 163–181.

Hsu, J.R.C., Uda, T., Silvester, R., (2000). Shoreline protection method – Japanese experience. in Herbich, J.B. (ed.), *Handbook of Coastal Engineering*, Chapter 9, McGraw Hill, 9.1–9.77.

Klein., A.H.F., Vargas, a., Raabe, A.L.A., Hsu, J.R.C., (2023). Visual assessment of bayed beach stability with computer software. Computers & Geomechanics, 29, 1249–1257.

Uda, T., (2010). *Japan's Beach Erosion: Reality and Future Measures*. Advanced Series on Ocean Eng., v. 31, World Scientific, 418 pp.

---

## Referee Comment (RC2)

This study addresses the severe beach erosion at Wolcheon Beach following the Samcheok LNG terminal's large-scale reclamation project. The authors use a shoreline change model, validated with satellite imagery, to analyze the effects of altered littoral drift (LST). The research is highly relevant given the increasing global issue of coastal erosion due to anthropogenic activities and climate change. The study provides valuable insights into predictive modeling and potential mitigation strategies. However, while the study presents strong empirical evidence, some aspects require further clarification, particularly regarding model assumptions, validation methodology, and the applicability of proposed mitigation measures. Below is a detailed critique.

Strengths of the Study

Timely and Relevant Topic

- Coastal erosion due to large-scale infrastructure projects is a pressing issue, and the study highlights an extreme case with real-world implications.
- The integration of satellite data and numerical modeling is commendable, as it allows for a robust spatiotemporal analysis.

Methodological Rigor

- The study effectively uses Google Earth Engine for satellite-based shoreline detection, a reliable method that enhances the spatial resolution of shoreline change assessment.
- The use of the Parabolic Bay Shape Equation (PBSE) to propose mitigation measures (e.g., groins) is methodologically sound and aligns with coastal engineering principles.

Clear Identification of Impacts

- The analysis clearly demonstrates the severe impacts of the Samcheok LNG reclamation on Wolcheon Beach, substantiating claims with quantitative LST analysis.
- The discussion on wave diffraction effects and their role in exacerbating LST-induced erosion is insightful and well-supported by existing literature.

Oversimplification of Sediment Transport Processes

- The one-line shoreline change model assumes uniform longshore transport but does not account for cross-shore dynamics (e.g., storm-induced sediment suspension and offshore transport).
- While wave diffraction effects are discussed, the study lacks wave energy dissipation analysis, which could refine the understanding of sediment transport pathways.

  .

Insufficient Discussion of Seasonal and Climatic Variability

- The study acknowledges seasonal variations in erosion rates, but no specific meteorological events (e.g., storms, typhoons) are analyzed to determine their relative influence.
- The role of sea level rise (SLR) and climate-driven changes in wave energy is not addressed. Given the long-term relevance of coastal management, this omission limits the broader applicability of the study.

Mitigation Strategies Require Further Justification

- The proposed groin installation is based on the PBSE approach, which is widely used in coastal engineering. However:
  - The optimal groin spacing and expected sediment retention efficiency are not thoroughly quantified.
  - The authors should discuss potential adverse effects of groin structures, such as down-drift erosion or sediment starvation in adjacent coastal areas.
- Alternative mitigation measures (e.g., beach nourishment, submerged breakwaters) should be compared in terms of cost-effectiveness and environmental impact.

Recommendations for Improvement

Expand Discussion on Mitigation Strategies

- Justify groin placement and spacing with numerical simulations of sediment retention efficiency.
- Compare the effectiveness of groins vs. beach nourishment vs. submerged breakwaters in mitigating erosion at Wolcheon Beach.
- Discuss potential negative consequences of groin installation.

This study provides important insights into the consequences of large-scale coastal reclamation on sediment dynamics. The integration of satellite-based shoreline change detection with numerical modeling is a significant strength, and the proposed mitigation strategies are valuable for coastal engineers and policymakers.

However, to improve its impact and applicability, the study should:

I suggest replacing the adjective "catastrophic" with an equivalent, such as "substantial". I also suggest clarifying which variables contribute to the RMSE, which also assumes non-negligible values. Expand the discussion on the limitations of the method and discuss the uncertainty associated with the proposed solutions..

---

## Author Comment (AC1)

**Response to Reviewers' Comments On manuscript Number: NHESS-2024-176**

**Title: Catastrophic beach erosion induced by littoral drift on nearby beach after Samcheok LNG's massive coastal reclamation project**

**R#**: Reviewer number (1, 2); **C#**: Comment/Response number; **A**: Authors' response.

Line number in Marked copy of R1 manuscript: L#

Note: Words/phrases/sentences that represent the response to the Editor and Reviewers' comments are highlighted in BLUE/RED color in the revised manuscript, while those from our own revision are also typed in BLUE/RED.

A **Clean** copy is also produced, upon deleting the parts struck out and retaining only those newly added in **BLUE/RED**.

First of all, the author would like to express sincere thanks to the editor and two reviewers who reviewed this paper. The author responded to all comments individually in the following sections. Please refer to the line numbers in the two reviewers' comments (R#-C#) are information about the manuscript before correction and supplementation.

**Reviewer #1:**

**Reviewer #1: A. General comments**

R1-C0: Reviewer X has read the manuscript (MS) for "nhess-2024-176" (PDF).

This MS reports the analysis of 10 satellite images of beach changes (2021.03.13 to 2022.12.12) at Samcheok LNG terminal, verification of beach changes using numerical calculations (with longshore transport, LST) and empirical bay shape model (PBSE – Hsu and Evans, 1989 with software MeePaSoL – Lim et al., 2021), and suggestion applying groins to protect/prevent beach from erosion prior to the construction of a large-scale coastal development project.

Overall, the layout of the presentation (e.g., sections and sub-sections) is in good order and all results are valuable for coastal managers, planners, and engineering consultants on a large-scale coastal project, from which beach erosion and shoreline rotation could occur, during or after the construction, arising from changes of nearshore wave field.

The title of the MS "Catastrophic beach erosion induced by littoral drift on nearly beach after Samcheok LNG's massive coastal reclamation project" is catchy and attractive, especially in using the words "catastrophic" and "massive". In reality, beach erosion downdrift of a harbor breakwater is the norm, which has been known for decades and also documented (e.g., Hsu et al., 1993; Hsu et al., 2000; Uda, 2010). Therefore, the state of beach erosion at the scale of about 40 m may be referred as "severe", which can be expected, instead of "catastrophic".

In addition, the beach that suffered erosion is not at "nearby", but more specifically at "downdrift", or "immediately downdrift". The main cause to erosion at Samcheok was not directly "induced by littoral drift", instead, it was associated with "wave-induced nearshore circulation that transported sediment within the shadow zone of the diffracted waves in the lee of a harbor breakwater or detached breakwater" (e.g., Gourlay, 1974; 1981). In addition, the preventive strategy using groins to control erosion can be found in Hsu at al. (2000), who reported examples of Japanese experience in the 1970-80s.

**A1-C0:** Thank you for your positive comments on this paper. Following your comments and combining it with the suggestions from other reviewers, we have revised the title to "Severe

beach erosion induced by shoreline deformation after a large-scale reclamation project for Samcheok LNG terminal in Korea."

Regarding your comments on "induced by littoral drift", since the nearshore circulation caused by diffraction ultimately leads to littoral drift and results in erosion, we consider these expressions to be similar. For further clarification, we have also added your comment to the revised version. Additionally, we have added the reference you mentioned, Hsu et al. (2000), to the discussion section as follows.

**Reviewer #1: B. Specific Comments**

R1-C1: (L1-2) Title of the paper: Suggestion --- May be modified as

"Severe beach erosion induced by shoreline rotation after a large-scale reclamation project for Samcheok LNG terminal in Korea"?

A1-C1: We have revised the title as you suggested:

Severe beach erosion induced by shoreline deformation after a large-scale reclamation project for Samcheok LNG terminal in Korea

**R1-C2:** (L30-31) "These marine life habitats contribute to water pollution, significantly affecting marine ecosystems and jeopardizing coastal infrastructures, housing, and facilities, ...": Is this statement correct?

**A1-C2:** The sentence in your comment was removed as it was deemed unnecessary and inappropriate to the topic of the paper, although it can be confirmed in the cited reference.

**R1-C3:** (L43-45) "... the double headland method..., it resulted in excessive diffracted waves...": What is the "excessive" in diffracted waves?

**A1-C3:** The expression "excessive diffraction waves" was deemed inappropriate and changed to " excessive beach distribution by diffraction waves ".

**R1-C4:** (L47-48) "Changing wave fields caused by ports and coastal structures influence coastal sediment transport, leading to shoreline alterations and erosion.":

What was the mechanism behind this phenomenon?

**A1-C4:** Beach erosion caused by coastal structures is a topic covered in many literatures. However, there were some awkward parts in the expression in the sentence, so we modified it as follows. We also cited a paper that covered this mechanism.

- Wave deformation caused by coastal structures mainly influences the longshore sediment transport, leading to shoreline reshaping (Lim et al., 2021).

**R1-C5:** (L52-53) "... LST is considered more influential than episodic cross-shore sediment transport in driving significant shoreline change over extended periods.":

Should we say that "LST" was the consequence of the change in nearshore wave field, but not the direct or primary cause to erosion at Samcheok?

(See the last paragraph under A. Specific Comments)

A1-C5: Based on your comment, we have replaced it with the following:

- LST, which is a consequence of the change in the nearshore wave field, is considered more influential than episodic cross-shore sediment transport in driving significant shoreline change over extended periods.

**R1-C6:** (L70-71) " ... significant wave diffraction from reclaimed revetment or breakwaters constructed outside ports.":

Why "significant wave diffraction"? What is the "reclaimed revetment" and "breakwaters constructed outside ports"?

A1-C6: To make it clearer, the sentence has been revised as follows:

- Although this model is widely used in engineering consulting, it underestimates results in scenarios with wave diffraction from large-scale coastal structures (Lee and Hsu, 2017).

**R1-C7:** (L124-127) "...The coastal waters of Samcheok...have high waves. ... Wolcheon Beach, the root mean square (RMS) wave height is estimated to be 1.14 m...":

Should we classify wave height in the order of 1.14 m as "high wave"? (See also Fig. 3a for wave height)

**A1-C7:** The expression "high wave" was removed because it was judged not to meet global classifications.

**R1-C8:** (L128-137) "Figure 4...the resulting rose diagram of LST components (green: north, orange: south). ... The dominant direction of wave incidence for the static equilibrium of Wolcheon Beach was found to be 34.2N..." and Fig. 4:

Given the dominate wave direction was 34.2N, then the average straight shoreline in equilibrium at Wolcheon Beach before 2011.03.13 had inclination from 124.2N to 304.2N (Left panel of Fig. 2 in the MS):

(calculation: N34.20 + 900 = N124.20; N34.20 + 2700 = N304.20).

Consequently, the LST direction in summer (South to North) should be within 90 - 124N and in winter (North to South) within 304 - 338N, respectively.

However, the directions indicated above differ from the rose diagram for LST shown in Fig. 4 in the MS, which was centered around 230N in summer (Green, South to North) and centered around 180N in winter (Orange, North to South), respectively, all showing seaward LST.

**PLEASE CONFIRM/VERIFY!**

**A1-C8:** The littoral rose shown in Figure 4 is different in degree from that shown in the wave rose. A simple calculation from the average value of the wave direction is the result extracted from the wave rose. However, if littoral rose is expressed in the same way as wave rose, the direction of littoral drift may be reversed, which may confuse readers.

As per your comment, there may be a misunderstanding that littoral rose is related to the angle with respect to the wave direction, so I added the following sentence:

- The angle shown in Figure 4 represents the wave direction with respect to the wave rose. Figure 4 also shows the littoral rose according to wave rose, drawn symmetrically around the vertical line of the dominant direction (N124.2° - N304.2°).

**R1-C9:** (L135) Figure 3: "Annual" values for mean wave height, wave period, and wave direction" are meaningless, because they do not represent the seasonal variations (summer and winter), especially during the period of investigation between 2011.03.1 and 2012.12.12.

**A1-C9:** The annual mean value shown in Figure 3 is significant in that the wave climate has not changed much over a long period of time. Wave climate is the first thing to consider when analyzing the causes of beach erosion. However, according to Figure 3, the wave climate in Samcheok did not change significantly, so we can find out that severe beach erosion occurred due to other factors. We have added sentences for this explanation as follows:

- Figure 3 shows that wave climate, one of the main causes of beach erosion, has not changed significantly in Samcheok.

R1-C10: (L218-222, Figure 9) "...the downdrift control point X ...":

Should point X be behind the initial shoreline in Fig. 4 after beach erosion?

**A1-C10:** The definition sketch shows that the beach is open beyond the downdrift point. Therefore, unlike a closed cell, it may not be located behind the initial shoreline because of the continuous sediment supply.

**R1-C11:** (L259) "..., subjected to LST towards the estuary of the Gagok Creek due to diffraction waves.": Yes, but more explicitly, due to nearshore current circulation induced by diffracted waves, which carry sediment .....?

A1-C11: We have revised the expression as you suggested.

**R1-C12:** (L265-266) "...shoreline deformation began on August 17, 2021...": This occurred as LST moving within 90 – 112N or 90 – 124N ? (See also (L128 – 137) above)

**A1-C12:** As previously answered in **A1-C8**, this paper analyzes the severe beach erosion caused by the Samcheok LNG development rather than the shoreline deformation resulting from the change in wave climate. Therefore, we simulate a numerical model using the annual mean wave as input and analyze the cause of severe beach erosion.

**R1-C13:** (L267-270) "This is because shoreline change...lasts for long periods due to the LST by the oblique inflow of waves under ordinary wave conditions with a low wave height rather than the deformation by high waves.":

Wordy sentence. Please revise!!

**A1-C13:** It has been revised to make sentences clearer by removing unnecessary and redundant expressions.

**R1-C14:** (L321-326 and Figure 16) "Figure 16 compares the LST rate vectors... The numerical results show a consistent LST vector pattern towards the LNG terminal revetment except for the initial results, but the results obtained from satellite images do not always show a vector towards the LNG terminal revetment due to the effect of transient high wave inflow.":

Was this true (effect of transient high wave inflow)? where the high waves came?

In winter, the entire Wolcheon Beach might be covered within the shadow zone induced by the diffracted waves (please see the Samcheok LNG 20211023.jpg below; or Fig. 2b in MS), whereas in summer, oblique wave reflection from or along part of the slightly curved revetment

wall could form a short-crested wave system that transported sediment out and away from the revetment toward the beach?

**A1-C14:** In the case of high waves (including not only extreme surge waves from the NE but also larger-than-usual easterly waves exceeding 2–3 meters from E), the deep water in front of the newly constructed revetment likely resulted in a significant influence of wave reflection, as observed in satellite imagery.

Therefore, the following explanation has been added:

- The most reasonable cause of this phenomenon is that, due to the deep-water construction of the LNG terminal revetment, oblique wave reflection from or along part of the slightly curved revetment wall could have formed a short-crested wave system, transporting sediment away from the revetment and toward the beach.

**R1-C15:** (L327-330) "Figure 17 compares LST vectors with each other on average over the entire analysis period (from March 13, 2012 to Aug. 6, 2012)... ":

The term "on average" can only compare the end results, not the transient temporal process of this investigation during the 1.5 years, thus invalidating the purpose of comparation! Therefore, Figure 17 does not support the comparation it intended to serve.

In addition, the "entire" period was from "2021", not "2022"! A typo?

**A1-C15:** The expression means the average over the monitoring period from remote sensing. However, as per your comment, "on average" may be misleading so we have removed it.

In addition, we have revised to the entire period, "2011".

**R1-C16:** (L338-343; Figure 18 on L347-349) "Figure 18 compares the average magnitude of the LST vectors from each end grid and the cumulative amount of longshore sediment towards the LNG revetment. The positive number... towards the LNG...negative number towards the opposite direction of the LNG revetment. ... showed severe undulation... by smoothing through the back-and-front values... a fairly similar trend...":

(From Figure 18) Can authors explain the reason why LST obtained from satellite images decreased during the period of 255th-290th days and 305th-330th days, respectively, and LST away (negative numbers) from the revetment during 280th-295th days and 335th-370th days, respectively?

**A1-C16:** Landsat-7 has a resolution of 30 m, which includes errors in the monitoring results presented in this paper. Therefore, the LST extracted from satellite images in Figure 18 includes negative values, unlike the numerical results. To compensate for this, the smoothed satellite image results have also been included in Figure 18.

Therefore, the relevant content has been added as follows:

- Landsat-7 has a resolution of 30 m, which includes errors in the monitoring results presented in this paper. Therefore, the LST extracted from satellite images in Figure 18 includes negative values, unlike the numerical results.

**R1-C17:** (L351-355) "When the reclamation project was planned, action was not taken owing to the absence of means with which to predict such large-scale erosion in advance. ...assessing

the impact of the construction... the rotation of the shoreline... by applying PBSE... groins can serve as representative coastal structures for LST control.":

Yes, the statement is correct. Additional notes of "shoreline reshaping" or "rotation of shoreline" can be found in Klein et al. (2023).

Groin system had been installed on Oarai Harbor and Iwafune Harbor in Japan to control erosion in the 1980s (see Hsu et al., 1993; 2000).

At moderate length (protrusion), groins may be effective to mitigate local erosion, but cannot modify the curvature of an embayed shoreline given by the PBSE.

**A1-C17:** Thank you for your comment. We have added appropriate references to these sentences based on your comments.

**R1-C18:** (L395-398, Conclusions) "... When the reclamation project at the Samcheok LNG terminal was planned about 10 years ago there was no adequate means to predict such large-scale erosion in advance. Therefore, if numerical predictions like this study are carried out, various countermeasures are possible. ... so installing groin in advance can most effectively reduce erosion. Applying PBSE...": Use "were"?

**Agreed!**

Since the early 1990s, there are abundant of helpful knowledge available for shoreline changes at downdrift of harbors in Japan (see Uda, 2010), but perhaps none of the members on a planning committee (from government agency, academic, engineering consults, and NGO body) in any country outside Japan have the up-to-date knowledge to deal with this type of beach erosion problem (with shoreline rotation from straight to embayed), so leading to do- nothing or construction of conventional hard structures.

Without a preventive strategy proposed at the planning stage, such as the one outlined in the MS, a large sum of taxpayer's funds will be wasted on managing the beach erosion downdrift of a harbor following a large coastal project.

Ironically, similar scenario of beach erosion has occurred in many countries, rich or poor, developed or under-developed.

**A1-C18:** Thank you for your comment. We have added appropriate contents and references to these sentences based on your comments in Conclusions.

**Reviewer #1: C. Technical Corrections**

Language editing is strongly recommended for the entire MS to improve the readability and the overall quality of the paper, especially on the use of "determiner (a/an/the/this etc.) and revise the wordy sentences throughout the entire MS.

**R1-C19:** (L100) "... Wolcheon Beach which mainly suffered erosion damage...": Wordy sentence?

A1-C19: The sentence was too long as your comment, so we split it into two sentences.

**R1-C20: (L111) "... a trade port equipped with a 1,800 m breakwater,": Why 'equipped"?**

A1-C20: We have deleted "equipped," as it is an unnecessary expression.

R1-C21: (L113) " ... when public water reclamation began.": What is this?

**A1-C21:** We have deleted "when public water reclamation began." as it is an unnecessary expression.

**R1-C22:** (L124-125) "The coastal waters of Samcheok...are deep (maximum depth of 3,000 m or higher; average depth of 1,300 or less)":

Are the numbers (depth) correct? Use word "more"? Please note that the edge of the continental shelf has an average depth about 150 – 200 m!

**A1-C22:** We have used the word "more" as your comment. The depth referred to here is the depth at the Samcheok LNG terminal. To make the subject clearer, "LNG terminal" has been added to the sentence.

**R1-C23:** (L139-140) " ... in Samcheok coastal waters. The waters are significantly affected by the flow of waves rather than the tide.": Why "significantly" and "the flow of waves"?

A1-C23: To make it clearer, the sentence has been revised as follows:

- Due to the small tidal range, the beach are usually affected by the wave-induced currents rather than the tide.

**R1-C24:** (150-151) "the occurrence of a coastal erosion problem that considerably threatens the lives, properties, and livelihoods....": Wordy sentence!

**A1-C24:** The sentence has been removed as it was unnecessary for the development of the paper.

R1-C25: (L157) "Towing to the construction...": Typo error?

A1-C25: The sentence has been revised to "after the construction".

**R1-C1:** (L160-162) "Owing to the absence of the data... gravity s = 2.65, which were obtained from the sand sample collected from the sandy beach located in the sane watershed system were applied.":

Wordy sentence!

A1-C26: To make it clearer, the sentence has been revised as follows:

- Owing to the absence of the data in the target area, porosity p = 0.42 and sand gravity weight s = 2.65 are applied, as these are representative values in South Korea.

**R1-C27:** (L176) "... as Liu and Jezek (2004) pointed out a long time ago, this often still requires rigorous....": Poor sentence!

A1-C27: To make it clearer, the sentence has been revised as follows:

- However, as Liu and Jezek (2004) pointed out long ago, extracting the shoreline still often requires rigorous terrain correction, geocoding, and radiometric correction and balancing.

**R1-C28:** (L178) "However, as shown in the figure, However, as you can see in the figure, …": Careless and rough without checking!

A1-C28: We have removed redundant expressions.

R1-C29: (L321) "...two temporally adjacent aerial photographs...": Use "consecutive"?

A1-C29: We have revised to "consecutive satellite images".

**R1-C30:** (L396-397) "...if numerical predictions like this study are1 carried out, various2 countermeasures are3 possible.":

1 Use "were"? 2Use "proper" or "effective"? 3Use "could be"?

A1-C30: Thank you for your comment. We have made revisions based on your comment.

**R1-C31: D. Additional References other than those already included in the MS**

Gourlay, M.R., (1974). Wave set-up and wave generated currents in the lee of a breakwater or headlands. Proc. 14th Inter. Conf. Coastal Eng., pp. 1976–1987.

Gourlay, M.R., (1981). Beach processes in the vicinity of offshore breakwaters. Proc. 5th Aust. Conf. Coastal Eng., pp.129–134.

Hsu, J.R.C., Uda, T., Silvester, R., (1993). Beaches downcoast of harbours in bays. Coastal Eng., 19, 163–181.

Hsu, J.R.C., Uda, T., Silvester, R., (2000). Shoreline protection method – Japanese experience. in Herbich, J.B. (ed.), Handbook of Coastal Engineering, Chapter 9, McGraw Hill, 9.1–9.77.

Klein., A.H.F., Vargas, a., Raabe, A.L.A., Hsu, J.R.C., (2023). Visual assessment of bayed beach stability with computer software. Computers & Geomechanics, 29, 1249–1257.

Uda, T., (2010). Japan's Beach Erosion: Reality and Future Measures. Advanced Series on Ocean Eng., v. 31, World Scientific, 418 pp.

**A1-C31:** Thank you for your comment. We have added appropriate references to these sentences based on your comments.

---

## Author Comment (AC2)

**Response to Reviewers' Comments**
On manuscript Number: **NHESS-2024-176**

Title: Catastrophic beach erosion induced by littoral drift on nearby beach after
Samcheok LNG's massive coastal reclamation project

**R#**: Reviewer number (1, 2); **C#**: Comment/Response number; **A**: Authors' response.

Line number in **Marked** copy of **R1** manuscript: **L#**

Note: Words/phrases/sentences that represent the response to the Editor and
Reviewers' comments are highlighted in BLUE/RED color in the revised
manuscript, while those from our own revision are also typed in BLUE/RED.

A **Clean** copy is also produced, upon deleting the parts struck out and retaining only
those newly added in BLUE/RED.

First of all, the author would like to express sincere thanks to the editor and two reviewers who
reviewed this paper. The author responded to all comments individually in the following
sections. Please refer to the line numbers in the two reviewers' comments (R#-C#) are
information about the manuscript before correction and supplementation.

**Reviewer #2:**

**Reviewer #2: A. General Comments G**

This study addresses the severe beach erosion at Wolcheon Beach following the Samcheok
LNG terminal's large-scale reclamation project. The authors use a shoreline change model,
validated with satellite imagery, to analyze the effects of altered littoral drift (LST). The research
is highly relevant given the increasing global issue of coastal erosion due to anthropogenic
activities and climate change. The study provides valuable insights into predictive modeling and
potential mitigation strategies. However, while the study presents strong empirical evidence,
some aspects require further clarification, particularly regarding model assumptions, validation
methodology, and the applicability of proposed mitigation measures. Below is a detailed
critique.

**R2-C1:** Strengths of the Study Timely and Relevant Topic

- Coastal erosion due to large-scale infrastructure projects is a pressing issue, and the
  study highlights an extreme case with real-world implications.

- The integration of satellite data and numerical modeling is commendable, as it allows
  for a robust spatiotemporal analysis.

**A2-C1:** Thank you for your comments.

**R2-C2:** Methodological Rigor

- The study effectively uses Google Earth Engine for satellite-based shoreline detection,
  a reliable method that enhances the spatial resolution of shoreline change
  assessment.

- The use of the Parabolic Bay Shape Equation (PBSE) to propose mitigation measures
  (e.g., groins) is methodologically sound and aligns with coastal engineering principles.

**A2-C2:** Thank you for your comments.

**R2-C3:** Clear Identification of Impacts

- The analysis clearly demonstrates the severe impacts of the Samcheok LNG reclamation on Wolcheon Beach, substantiating claims with quantitative LST analysis.

- The discussion on wave diffraction effects and their role in exacerbating LST-induced erosion is insightful and well-supported by existing literature.

**A2-C3:** Thank you for your comments.

**R2-C4:** Oversimplification of Sediment Transport Processes

- The one-line shoreline change model assumes uniform longshore transport but does not account for cross-shore dynamics (e.g., storm-induced sediment suspension and offshore transport).

- While wave diffraction effects are discussed, the study lacks wave energy dissipation analysis, which could refine the understanding of sediment transport pathways.

**A2-C4:** As mentioned in Section 5.2, this study focuses more on shoreline reshaping with normal waves rather than coastal retreat caused by storm waves. This is because severe erosion damage at Wolcheon Beach is analyzed to have been primarily caused by the Samcheok LNG terminal. Furthermore, the annual variation in wave climate, as shown in Figure 3, appears to be minimal. This implies that a rough analysis can be conducted by simply estimating longshore sediment transport (LST) based on the average wave climate. For reference, the primary input for estimating LST indirectly considers energy suspension through breaking waves. The comments you provided have been additionally described as limitations in the Discussion section.

**R2-C5:** Insufficient Discussion of Seasonal and Climatic Variability

- The study acknowledges seasonal variations in erosion rates, but no specific meteorological events (e.g., storms, typhoons) are analyzed to determine their relative influence.

- The role of sea level rise (SLR) and climate-driven changes in wave energy is not addressed. Given the long-term relevance of coastal management, this omission limits the broader applicability of the study.

**A2-C5:** As explained in detail in **R2-C4**, this study focuses on analyzing the causes of erosion due to the Samcheok LNG terminal. Therefore, factors other than littoral drift were not analyzed in detail. However, recognizing the importance of these factors, we have elaborated on these limitations in the Discussion section.

**R2-C6:** Mitigation Strategies Require Further Justification

- The proposed groin installation is based on the PBSE approach, which is widely used in coastal engineering. However:

  - The optimal groin spacing and expected sediment retention efficiency are not thoroughly quantified.

  - The authors should discuss potential adverse effects of groin structures, such as down-drift erosion or sediment starvation in adjacent coastal areas.

- Alternative mitigation measures (e.g., beach nourishment, submerged breakwaters) should be compared in terms of cost-effectiveness and environmental impact.

**A2-C6:** Large-scale coastal development near shorelines can cause significant topographical changes to adjacent coastal areas. In many cases, economic priorities take precedence, making it impossible to halt development even in coastal areas with high conservation value. Therefore, this study proposes a solution using a hard engineering method, which, despite its drawbacks, is the most direct and effective approach to preventing critical topographical changes and sand loss.

Furthermore, in hard engineering, beach nourishment is unnecessary, as the sand that needs to be preserved already exists. Among the various functions of coastal structures, a groin has been suggested in the discussion as a means to mitigate sand loss caused by littoral drift. Additionally, the considerations of cost-effectiveness and environmental impact are regarded as separate and complex topics that fall beyond the scope of this study.

These details have been further explained in Section 6.2 of the discussion.

**R2-C7:** Recommendations for Improvement

Expand Discussion on Mitigation Strategies

- Justify groin placement and spacing with numerical simulations of sediment retention efficiency.

- Compare the effectiveness of groins vs. beach nourishment vs. submerged breakwaters in mitigating erosion at Wolcheon Beach.

- Discuss potential negative consequences of groin installation.

**A2-C7:** As per your advice, we have added a detailed discussion on groins to mitigate severe erosion damage caused by the installation of the Samcheok LNG terminal at Wonpyeong Beach. In particular, numerical results are presented for cases where submerged breakwaters (considering their locations and number) were installed to mitigate erosion damage in areas other than the estuary. Additionally, it was discussed that if the groin's protrusion length is too large, it may cause negative effects due to additional wave deformation.

Regarding the efficiency of each method, additional discussions were made on whether groins are more appropriate than beach nourishment, as detailed in **A2-C6**. Furthermore, it was determined that submerged breakwaters are not a suitable method as they do not block littoral drift.

**R2-C8:** This study provides important insights into the consequences of large-scale coastal reclamation on sediment dynamics. The integration of satellite-based shoreline change detection with numerical modeling is a significant strength, and the proposed mitigation strategies are valuable for coastal engineers and policymakers.

However, to improve its impact and applicability, the study should:

I suggest replacing the adjective "catastrophic" with an equivalent, such as "substantial". I also suggest clarifying which variables contribute to the RMSE, which also assumes non-negligible values. Expand the discussion on the limitations of the method and discuss the uncertainty associated with the proposed solutions.

**A2-C8:** Based on the comment from the two reviewers, the title of this paper has been revised as follows:

**Severe beach erosion induced by shoreline deformation after a large-scale reclamation project for Samcheok LNG terminal in Korea**

Additionally, considering the various suggestions made by Reviewer 2, a section explaining the limitations of this study has been added to the Discussion section.

---

## Author Response (AR2)

**Response to Reviewers' Comments**
**On manuscript Number: **NHESS-2024-176**

Title: Severe beach erosion induced by shoreline deformation after a large-scale reclamation project for Samcheok LNG terminal in Korea

Note: Words/phrases/sentences that represent the response to the Editor and Reviewers' comments are highlighted in BLUE/RED color in the revised manuscript, while those from our own revision are also typed in BLUE/RED.

A **Clean** copy is also produced, upon deleting the parts struck out and retaining only those newly added in BLUE/RED.

First of all, the authors sincerely thanks the editor and both reviewers for their valuable time and insightful comments, which have greatly contributed to improving the manuscript. All comments have been addressed in detail in the responses below.

**Editor:**

**Comment:** Considering the answers of the reviewers, which are both positive on the publication of this manuscript, it could be accepted after the residual specific change recommended by Reviewer#1.

Further, I suggest that in the final sentence (line 444-445) "would not have occurred" is replaced with "could have been avoided", as the implementation of the mitigation actions should not be given for granted in spite of the evidence suggested by this study.

**Response:** Thank you very much for your thoughtful suggestion. As advised, we have revised the sentence by replacing *"would not have occurred"* with *"could have been avoided"* to better reflect the conditional nature of implementing mitigation actions, as appropriately highlighted in your comment.

Additionally, we have addressed the minor revision requested by Reviewer #1, as detailed below.

**Reviewer #1:**

**Comment:** Please clarify the extend of water depth at the outer breakwater for the LNG terminal (10 or 15 or 20 m?) and the design wave conditions for the breakwater (wave height and period); instead of vaguely stating (Line 133) that "The coastal waters near the Samcheok LNG terminal, where Wolcheon Beach is located are deep and subjected to high wave energy".

**Response:** Thank you for your helpful comment. We have revised the text to clarify the water depth and wave conditions near the Samcheok LNG terminal as follows:

- The coastal waters near the Samcheok LNG terminal, where Wolcheon Beach is located, have depths ranging from approximately 20 to 30 m. These waters are exposed to moderate to high wave energy conditions, with a mean annual significant wave height between approximately 1.04 and 1.24 m.

**Reviewer #2:**

**Comment:** The authors have responded satisfactorily to the first report. As far as I am concerned, the manuscript can be accepted for publication.

**Response:** The authors would like to express sincere appreciation to Reviewer #2 for the positive feedback and recommendation for publication.